# Influence of secondary ice formation on tropical deep convective clouds simulated by the Unified Model

Mengyu Sun[1], Paul J. Connolly[1], Paul R. Field[2,3], Declan L. Finney[3,4], Alan M. Blyth[4,3]

[1] Department of Earth and Environmental Sciences, University of Manchester, Manchester, UK
[2] Met Office, Exeter, UK
[3] Institute for Climate and Atmospheric Science, School of Earth and Environment, University of Leeds, Leeds, UK
[4] National Centre for Atmospheric Science, Leeds, UK

*Correspondence to*: Mengyu Sun (mengyu.sun@manchester.ac.uk)

## Abstract

Secondary ice production (SIP) plays an important role in tropical deep convection. This study implements multiple SIP mechanisms, including droplet fragmentation and ice–ice collisional breakup, into the CASIM microphysics scheme of the UK Met Office Unified Model, and evaluates their impacts through a real-case simulation of a Hector thunderstorm. SIP enhances ice number concentration in upper cloud layers, with values up to 3 orders of magnitude higher than the no-SIP case, particularly above -10 °C. Ice water content (IWC) increases by a factor of 3–5 in the anvil region, contributing to more extensive upper-level cloud coverage. These microphysical changes reduce outgoing longwave radiation (OLR) by ~3.2 W m$^{-2}$ (1.3%) and increase outgoing shortwave radiation (OSR) by ~4.5 W m$^{-2}$ (1.8%) over a 6-hour analysis period and a 110 km × 110 km domain. SIP modifies precipitation spatially, yielding a more localized, compact rainfall pattern near the convective core, while reducing domain-averaged precipitation by ~8%. Peak rainfall rates remain only slightly affected, consistent with the minor changes (< 1 m s$^{-1}$) in maximum updraft velocity. Among the tested mechanisms, ice–ice collisional breakup shows negligible impact on simulated ice concentration, consistent with limited graupel-involved collision energetics under warm profiles. Ensemble experiments confirm that these effects are robust and exceed the influence of meteorological variability. These results highlight the importance of representing SIP processes in cloud-resolving models of tropical convection and accounting for their environmental dependence.

# 1 Introduction

The formation of ice particles in mixed-phase and deep convective clouds plays a central role in cloud microphysics, precipitation processes, and cloud–radiation interactions. Observational studies have consistently shown that the number concentrations of ice particles in clouds often exceed that explained by primary ice nucleation alone (Hallett et al., 1978; Hobbs and Rangno, 1985; Cantrell and Heymsfield, 2005; Korolev et al., 2022). This discrepancy has been widely attributed to secondary ice production (SIP) processes (Field et al., 2017; Korolev et al., 2020), which generate additional ice particles from existing hydrometeors through mechanisms such as rime splintering (Hallett and Mossop, 1974), droplet shattering during freezing (Dye and Hobbs, 1968; Keinert et al., 2020; James et al., 2021), and ice–ice collisional breakup (Vardiman, 1978; Takahashi et al., 1995; Grzegorczyk et al., 2023; Gautam et al., 2024; Yadav et al., 2025). SIP primarily occurs in the mixed-phase region of clouds. It can significantly enhance upper-level ice crystal concentrations, alter the hydrometeor size distribution, and affect cloud lifetime, precipitation efficiency, and radiative fluxes at the top of the atmosphere (Lohmann, 2006; Kudzotsa et al., 2016; Han et al., 2024; Waman et al., 2025). Understanding and quantifying the effects of SIP using cloud-resolving models is crucial for improving the simulation of convective cloud systems and their feedback in the climate system.

Despite significant progress in identifying and parameterizing SIP processes the application of parameterizations across different cloud regimes remains uncertain. Most existing SIP schemes were originally developed based on midlatitude stratiform or polar mixed-phase clouds, where environmental conditions such as temperature, liquid water content, and hydrometeor types differ markedly from those in tropical deep convection (Phillips et al., 2017a; Zhao et al., 2021). In deep tropical convection, warmer and more humid profiles, elevated freezing levels, and less abundant graupel may inhibit the activation of certain SIP mechanisms. For example, the classic rime-splintering mechanism (Hallett and Mossop, 1974) requires a narrow temperature window (-3 °C to -8 °C) and the co-existence of supercooled cloud droplets and graupel. These conditions may occur under favorable thermodynamic environments (Bazantay et al., 2025), but are not always present in tropical convective updrafts (Field et al., 2017; Huang et al., 2022). Moreover, recent laboratory studies found no experimental evidence of efficient rime-splintering under convective conditions (Seidel et al., 2024), suggesting that the relevance of this process and its parameterization remain uncertain. Other processes such as raindrop freezing fragmentation (Mode 1), drop–ice collisions with splashing/shedding (Mode 2), and ice–ice collisional breakup (BR) also exhibit strong environmental dependencies related to turbulence intensity, hydrometeor interactions,

and liquid water availability (Lauber et al., 2018; Phillips et al., 2018; Korolev et al., 2020). While all these mechanisms have been implemented in various model microphysics schemes, few studies have systematically compared them in a unified framework (e.g., Hawker et al., 2021a, 2021b). Consequently, their relative contributions and possible interactions remain poorly understood, limiting the physical realism and environmental adaptability of current SIP parameterizations (Han et al., 2024; Grzegorczyk et al., 2025a).

Beyond its role in modulating cloud microphysics, SIP may also have important impacts on precipitation structure and radiative fluxes, especially in tropical convective systems. By altering the number and type of ice-phase particles aloft, SIP can influence cloud development and precipitation formation (Qu et al., 2022; Waman et al., 2022). For instance, some studies have suggested that SIP enhances localized precipitation near convective cores (e.g., Sullivan et al., 2018), while others have reported reductions in domain-averaged rainfall linked to reduced warm-rain contributions (Han et al., 2024; Grzegorczyk et al., 2025b, c). However, these effects appear to be highly sensitive to the specific SIP mechanisms involved, and the underlying thermodynamic environment. Moreover, the importance of different SIP mechanisms may also be modulated by the availability and properties of ice-nucleating particles (INPs), which affect primary ice formation and the conditions favoring SIP activation (Hawker et al., 2021a, 2021b).

At the same time, SIP-induced changes to ice particle concentration and distribution, particularly in the upper-levels, can modify anvil cloud optical thickness, thereby influencing top-of-atmosphere radiative fluxes (Young et al., 2019; Zhao and Liu, 2021). Increased ice loading and the expansion of anvil cloud area may lower outgoing longwave radiation (OLR) due to colder and more extensive cloud tops, while enhanced shortwave reflectivity (i.e., outgoing shortwave radiation, OSR) may result from higher concentrations of small ice crystals (McKim et al., 2024; Finney et al., 2025). Despite these potential impacts, few modeling studies have explicitly evaluated how SIP mechanisms affect both precipitation and radiation in a coupled manner. Given the prominence of deep convection in the tropical energy budget and climate system, a more integrated assessment of SIP-driven changes to precipitation efficiency and radiative transfer is urgently needed.

To address these uncertainties, this study implements multiple SIP mechanisms, including droplet-freezing fragmentation (Mode 1 and Mode 2) and ice–ice collisional breakup, into the double-moment cloud microphysics scheme (CASIM) of the Unified Model (UM), which already includes rime splintering by default. A Hector-type tropical convective system (Crook, 2001; Connolly et al., 2006) is

simulated, and multi-source observational datasets are used to benchmark the model outputs.

We begin by examining how SIP modifies the storm's radiative and dynamical characteristics, including OLR, OSR, and upper-level anvil development. Next, we assess SIP-induced changes in the spatial distribution and intensity of precipitation. Finally, we analyze the microphysical pathways by comparing vertical profiles of ice-phase hydrometeors, enabling us to interpret the macro-scale responses in terms of specific SIP mechanisms. A set of sensitivity experiments is conducted to assess the effects of individual and combined SIP processes under tropical convective conditions. The model setup, experiment design, and evaluation metrics are described in the following sections.

## 2 Methodology

### 2.1 Model description

We use the Met Office Unified Model (UM) to conduct simulations of Hector thunderstorms over the Darwin region, during research as part of the ACTIVE campaign. The UM can be applied at a range of scales and geographical regions for weather forecasting.

The UM uses the ENDGame semi-Lagrangian dynamical formulation to solve the non-hydrostatic equations of motion (Wood et al., 2014). For regional-scale modeling at resolutions of order 1 km presented there is no parameterized convection. The SOCRATES scheme (Manners et al., 2023) is used for radiative transfer based on Edwards and Slingo (1996). Boundary layer turbulence is represented with a "blended", three-dimensional turbulent mixing scheme. This boundary-layer parameterization (Boutle et al., 2014; Bush et al., 2023) includes any non-local contribution from the scheme of Lock et al. (2000).

Cloud microphysics is described by the Cloud-AeroSol Interacting Microphysics (CASIM) module (Miltenberger et al., 2020; Field et al., 2023). CASIM is a multi-moment bulk scheme, which in this study is configured to be double moment. It can predict the mass and number of five hydrometeors species (cloud droplets, raindrops, ice crystals, graupel, and snow) prognostically. The size distribution of different species is assumed to be a generalized gamma distribution. Specific hydrometeor parameters including the terminal fall speed velocities, shape and mass dimension are shown in Table A1 of Field et al. (2023). In our configuration, the ice-cloud optical properties used by SOCRATES (Manners et al., 2023) are diagnosed consistently from the CASIM-predicted ice water content and ice-phase number concentrations, so changes in ice amount and size distribution associated with different SIP mechanisms tend to be reflected in the simulated radiative fluxes. In CASIM, the cloud droplet number concentration ($N_d$) can be either diagnosed from background aerosol via explicit cloud condensation nuclei (CCN)

activation or prescribed as a fixed droplet number. To isolate and examine SIP-induced responses, we adopt a fixed $N_d=400$ cm$^{-3}$ in this study. This value falls within the CCN activation range reported for Hector storms in the pre-monsoon period (Connolly et al., 2013) and gives the best overall agreement with the available observations in this case (figures not shown). CASIM considers homogeneous freezing of cloud droplets, heterogeneous freezing, aggregation of ice crystals, sedimentation of ice-phase hydrometeors, and rime splintering. Homogeneous freezing of cloud droplets happens when temperatures is below the temperature threshold (-38 °C) and cloud water mass exceeds $10^{-6}$ g$\cdot$kg$^{-1}$, following Jeffery and Austin (1997). The parameterization of heterogeneous ice nucleation given by Cooper (1986) is used in this study. It is independent of the aerosol concentration and calculates the freezing rate based on temperature. Heterogeneous freezing of rain is represented following Bigg (1953). SIP from riming-splintering is parameterized following Hallett and Mossop (1974). We describe this together with other newly implemented SIP processes in Section 2.2.

## 2.2 Implementation of secondary ice production in the CASIM

In addition to the existing SIP process (i.e., riming splintering), we have implemented three new SIP mechanisms in the CASIM, including ice-ice collisional breakup, droplet shattering during symmetrical freezing, and droplet shattering during asymmetrical freezing, based on previous laboratory and theoretical research (Phillips et al., 2017b, 2018; James et al., 2021, 2023). The corresponding equations are reproduced below to explicitly show the parameterizations applied in our simulations.

### 2.2.1 Rime splintering

CASIM microphysics already includes the ice splinter production through riming (RS), which is known as the Hallett-Mossop process. The number and mass of secondary ice particles are produced during the accretion of water droplets by graupel and snow. The ice production rate is based on a triangular function (Hallet and Mossop,1974):

$$P_{ihal} = 3.5 \times 10^8 M_{I0}(P_{gacw} + P_{sacw})f_{RS}(T) \qquad (1)$$

where $P_{ihal}$ is the splinter production rate (units: number kg$^{-1}$ s$^{-1}$), $P_{gacw}$ and $P_{sacw}$ are the riming rate of cloud droplets by graupel and snow respectively, and $M_{I0}$ is the mass of each splinter. The temperature-dependent function, $f_{RS}(T)$, has a maximum value of 100% at $T = $ -5 °C and falls off linearly to zero at $T < $ -2.5 °C or $T > $ -7.5 °C. A maximum splinter production per rimed ice is set at $3.5 \times 10^8$ fragments per kilogram at -5 °C, based on laboratory experiments (Hallett and Mossop, 1974; Mossop, 1985).

### 2.2.2 Ice–ice collisional breakup

Parameterization of ice-ice collisional breakup (BR) is based on energy conservation principle (Phillips et al., 2017b), in which collision kinetic energy (CKE) is the fundamental governing variable of fragmentation in collisions of microphysical species (e.g., ice crystals, graupel, snow, or freezing drops). The production rate of ice particles number concentration ($\frac{\partial n_{ice}}{\partial t}|_{CB}$, units: number m$^{-3}$ s$^{-1}$) is calculated as follows:

$$\frac{\partial n_{ice}}{\partial t}\Big|_{CB} = \frac{\pi}{4}\, E\, \iint_0^\infty N_{CB}(D_1, D_2)\,(D_1 + D_2)^2\,|v(D_1) - v(D_2)|\, f(D_1)\, f(D_2)\; dD_1\, dD_2 \tag{2}$$

where E is the collisional efficiency (assumed constant), $D_1$ and $D_2$ represent the diameters of the colliding ice-phase particles (ice crystals, snow, or graupel), and $v(D_1)$ and $v(D_2)$ are the fall speeds of these particles with diameters $D_1$ and $D_2$. The functions $f(D_1)$ and $f(D_2)$ denote the size distributions, defined as $f(D_1) = \frac{dn_{ice}}{dD_1}$, and similarly for $f(D_2)$. $N_{CB}$ represents the breakup number of fragments per collision and is calculated as follows:

$$N_{CB} = \alpha A \left(1 - \exp\left\{-\left[\frac{CK_{0(CB)}}{\alpha A}\right]^\gamma\right\}\right) \tag{3}$$

in which $\alpha = \pi D^2$ is the equivalent spherical area of the colliding particle, $A$ is the number density of the breakable asperities in the contact region, $C$ is the asperity-fragility coefficient, and $\gamma$ is a parameter of riming intensity. $K_{0(CB)}$ represents the CKE between two colliding ice particles and is calculated as follows:

$$K_{0(CB)} = \frac{1}{2}\frac{m_1 m_2}{m_1 + m_2}(v_1 - v_2)^2 \tag{4}$$

where $m_1$ (and $m_2$), $v_1$ (and $v_2$) are the mass and fall speeds of both. Three broad types of collisions are categorized in Phillips et al. (2017b), in which Type I represents the collision of graupel with other graupel, type II is the collision of ice crystals or snow with graupel, and type III is the collision of crystals or snow with other crystals/snow. The parameters $A$, $C$ and $\gamma$ depend on the collisional type. Further details are provided in Table 1 and the Appendix of Phillips et al. (2017b). This parameterization is related to CKE, rimed fraction, temperature, and size of ice particles. For collisions without graupel (e.g., ice–ice and ice–snow), we prescribe a representative rime fraction of 0.1 based on ACTIVE observations; collisions involving graupel follow Phillips et al. (2017) Table 1 and are independent of rime fraction. In addition, numerical simulations indicate that snow–graupel collisions account for the majority of breakup fragments under convective conditions (Phillips et al., 2017b). The integral is evaluated online using a

 numerical integration routine within the CASIM scheme, and the collection efficiencies and terminal fall-speed parameters are given in Field et al. (2023), Appendix A, Tables A3 and A1, respectively.

**2.2.3 Droplet shattering during rain freezing**

Ice multiplication during fragmentation of freezing raindrops has also been implemented in the CASIM, using the parameterization proposed by Phillips et al. (2018). Two modes of droplet shattering were identified based on the relative weight of raindrops and ice particles.

In Mode 1 (M1), raindrops freeze due to immersion INPs and then shatter into smaller ice fragments. These drops are typically more massive than the ice particles they collide with. The predicted fragment numbers are significantly enhanced near -15 °C. By fitting the Lorentzian distribution to the laboratory data, the number of total ($N_{M1T}$) and large ($N_{M1L}$) fragments are given as:

$$N_{M1T} = F(D_R)\Omega(T)\left[\frac{\zeta\eta^2}{(T-T_0)^2+\eta^2} + \beta T\right] \tag{5}$$

$$N_{M1L} = \min\left\{F(D_R)\Omega(T)\left[\frac{\zeta_B\eta_B^2}{(T-T_{B0})^2+\eta_B^2}\right], N_T\right\} \tag{6}$$

where the parameters $\zeta$, $\eta$, $\beta$, $\zeta_B$, $T_0$, $T_{B0}$ are derived from Phillips et al. (2018). The onset functions for fragmentation are defined as $F(D_R) = \Delta_0^1(D_R, D_0, D_0 + \Delta D_R)$ and $\Omega(T) = \Delta_0^1(-T, -T_c, -T_c + \Delta T)$, where $F = 0$ for sizes below $D_0 = 50$ μm and $F = 1$ above $D_0 + \Delta D = 60$ μm; similarly, $\Omega = 0$ for $T > T_c = -3$°C and $\Omega = 1$ if $T < T_c - \Delta T = -6$ °C. Here, $D_R$ is the drop diameter just before freezing, and $T$ is the freezing temperature of the raindrop. More details are provided in Table B1 and Appendix B of Phillips et al. (2018).

The total rate of secondary ice production due to M1 ($\frac{\partial n_{ice}}{\partial t}|_{M1}$, units: number m$^{-3}$ s$^{-1}$) includes two contributions. The first term represents the freezing of supercooled raindrops via heterogeneous nucleation and the subsequent generation of fragments from each freezing event. The second term accounts for the collisional breakup between falling raindrops and less massive ice crystals, given as follows:

$$\frac{\partial n_{ice}}{\partial t}|_{M1,f} = \int_0^\infty (N_{M1T} + N_{M1L}) f(D_R) \, dD_R \tag{7}$$

$$\frac{\partial n_{ice}}{\partial t}|_{M1,c} = \frac{\pi}{4} E \int_0^{D_i=D_{i,thresh}} \int_0^\infty (N_{M1T} + N_{M1L})(D_R + D_i)^2 |v(D_R) - v(D_i)| f(D_R) f(D_i) \, dD_R \, dD_i \tag{8}$$

$$\frac{\partial n_{ice}}{\partial t}|_{M1} = \frac{\partial n_{ice}}{\partial t}|_{M1,f} + \frac{\partial n_{ice}}{\partial t}|_{M1,c} \tag{9}$$

where $D_R$ and $D_i$ are the diameters of raindrops and ice particles, $D_{i,thresh}$ denotes the diameter of an ice particle whose mass equals that of the colliding raindrop, $v(D_R)$ and $v(D_i)$ are the fall speeds, and $f(D_R)$

and $f(D_i)$ represent size distributions of freezing raindrops and ice particles, respectively. A corresponding equation for the secondary ice mass production rate has the same structure but integrates the fragment mass. These are calculated numerically in CASIM.

For Mode 2 (M2), a theoretical approach is taken to consider the collisions of supercooled raindrops with more massive ice. By assuming the fragmentation is controlled by the ratio of initial CKE and surface energy, the number of fragments generated due to M2 is given as follows:

$$N_{M2} = 3\Phi(T) \times [1 - f(T)] \times \max(DE - DE_{crit}, 0) \tag{10}$$

where $\Phi$ is the empirical fraction of the drop fragments from the splash contain ice. Following James et al. (2021), a value $\Phi = 0.3$ is used in this study. The function $f(T)$ represents the mass fraction of drop frozen and is given as:

$$f(T) = \frac{-c_w T}{L_f} \tag{11}$$

where $c_w = 4200 \, J \, kg^{-1} \, K^{-1}$ is the specific heat capacity of liquid water, and $L_f = 3.3 \times 10^5 \, J \, kg^{-1}$ is the specific latent heat of freezing. $DE$ is dimensionless energy and is calculated as:

$$DE = \frac{K_0}{S_e} \tag{12}$$

$$S_e = \gamma_{liq} \pi D_R^2 \tag{13}$$

where $K_0$ is the initial CKE defined in the preceding sections, $S_e$ is the surface energy, with $\gamma_{liq} = 0.073 \, J \, m^{-2}$ being the surface tension of liquid water, and $D_R$ the drop diameter. $DE_{crit}$ is the critical value of $DE$ for onset of splashing on impact and is set to be 0.2 (Phillips et al., 2018).

The corresponding secondary ice production rate due to M2 ($\frac{\partial n_{ice}}{\partial t}|_{M2}$, units: number $m^{-3} \, s^{-1}$) is calculated as:

$$\frac{\partial n_{ice}}{\partial t}|_{M2} = \frac{\pi}{4} E \int_{D_i = D_{i,thresh}}^{\infty} \int_{D_R = D_{R,thresh}}^{\infty} N_{M2} (D_R + D_i)^2 |v(D_R) - v(D_i)| f(D_R) f(D_i) \, dD_R \, dD_i \tag{14}$$

where $D_{R,thresh} = 0.15$ mm is the minimum raindrop diameter for Mode 2 (Phillips et al., 2017a). This integral is evaluated numerically using the CASIM (Field et al., 2023). It accounts for collisions between raindrops and ice particles, where the ice category includes cloud ice, snow, and graupel. All other variables have been defined above.

## 2.3 Observation and case description

In this study, we focus on the Hector thunderstorms observed during the Aerosol and Chemical Transport in tropIcal conVEction (ACTIVE) campaign (Vaughan et al., 2008). Hector is an isolated maritime continental thunderstorm that occurs regularly over the Tiwi Islands, north of Darwin, Australia

(Figure 1), during the pre-monsoon wet season. These tropical deep convective storms are primarily triggered by cold air pool interactions and the convergence of penetrating sea-breeze flows, making them some of the deepest convective systems observed globally (Connolly et al., 2013). The repeatable occurrence of Hector, featuring extensive mixed-phase cloud structure, supported by multi-platform observations from ACTIVE, provides a controlled setting to evaluate SIP processes. For this study, we selected the severe Hector thunderstorm that developed on 30 November 2005, coinciding with the early phase of the ACTIVE campaign. The 00 UTC Darwin sounding (Figure 2) indicates a low lifted condensation level (LCL) near 900 hPa and high convective available potential energy (CAPE) of $\sim 2590$ J kg$^{-1}$, with cloud tops extending above 200 hPa.

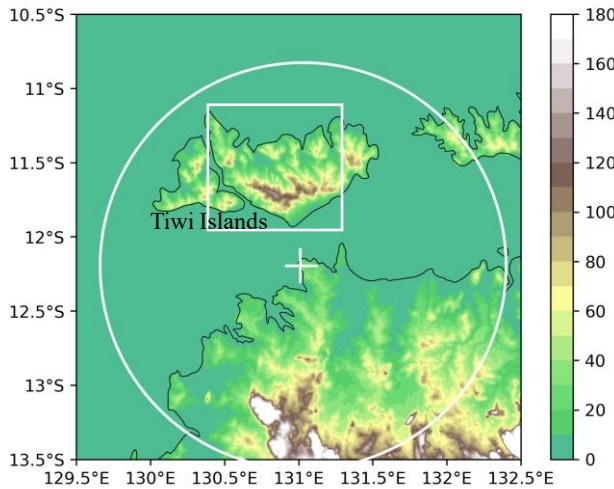

**Figure 1.** Overview of the model orography (terrain height, unit: m) and the domains used to generate radar and satellite datasets. The simulated area is a square domain centered at (12.4°S, 130.8°E) with a side length of 1350 km. In this panel, we only show the analysis domain for the radar data and simulated results (white box). The radar range is represented by the circle centered at (12.25°S, 131.05°E).

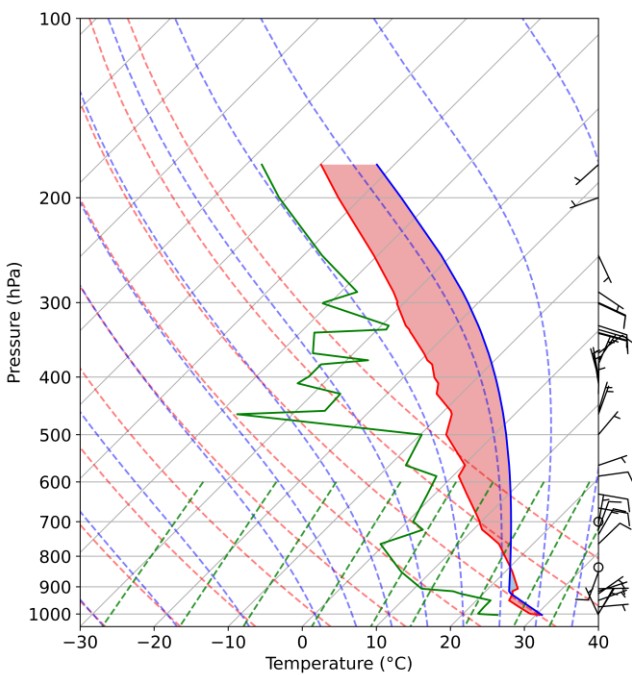

**Figure 2.** Sounding profiles from Darwin Airport station (12.42°S, 130.89°E) at 00 UTC on 01 December 2005. The red and green solid lines represent temperature and dew point, and the blue line represents the parcel lapse rate.

The radar reflectivity data were obtained from a C-band polarimetric (CPOL) radar, which was located at Gunn Point (12.25°S, 131.05°E) and updated every 10 min. Figure 1 illustrates its position relative to the Tiwi Islands, with the circle indicating its coverage range. The vertical and horizontal resolutions are 0.5 km and 2.5 km, respectively. To compare with the model, we utilized the "Statistical Coverage Product", which represents the time-height distribution of the fraction of a defined domain exceeding a threshold of 10 dBZ (May et al., 2009). This threshold was selected to capture the overall precipitation distribution, and the analysis domain is shown in Figure 1. The rainfall data at 10-min intervals was also derived from CPOL radar with a horizontal resolution of 1 km. CPOL-based quantitative precipitation estimates are considered reliable for tropical rainfall, with residual uncertainties arising from reflectivity/differential-reflectivity calibration and C-band attenuation (Louf et al., 2019; Jackson et al., 2021). The equivalent Outgoing Longwave Radiation (OLR) was calculated from the Multifunctional Transport Satellites (MTSAT) channel 1 brightness temperature, at 1.25 km horizontal and 30 min temporal resolution.

**2.4 Model configuration and sensitivity experiments**

The regional simulations presented have a horizontal grid spacing of 1.5 km (900 × 900 grid points), centered over the Darwin region (12.4°S, 130.8°E). There are 90 levels in the vertical stretched to 40 km

(52 levels below 10 km, 33 levels below 4 km, and 16 levels below 1 km). The simulations began at 00 UTC on 30 November 2005 and were integrated for 36 hr. The initial and lateral boundary conditions are from European Centre for Medium-range Weather Forecasts (ECMWF) reanalysis data at 6 hr intervals. The spin-up period is approximately 3 hr and the time step is 75 s. We utilize Shuttle Radar Topography Mission (SRTM) dataset as ancillary input to improve terrain representation. For analysis, we select an approximately $1° × 1°$ latitude-longitude domain over the Tiwi Islands (white box in Figure 1) to avoid analyzing data influenced by boundary effects.

A detailed description of the model sensitivity experiments and the included SIP mechanisms is provided in Table 1. The all-SIP experiment incorporates all SIP mechanisms, while additional experiments isolate specific combinations or exclude SIP entirely to evaluate their individual and collective effects. For example, the RS, RS+M1, RS+M2, and RS+BR experiments activate subsets of SIP processes, whereas the no-SIP configuration disables all SIP processes. These simulations aim to quantify how different SIP pathways influence cloud microphysics, convective dynamics, and precipitation. For each simulation, a time-lagged ensemble of four members with different initial conditions was performed. The simulations were initialized at 00, 06, 12, and 18 UTC on 30 November 2005 (hereafter referred to as T00, T06, T12, and T18), to ensure the resulting differences can be attributed to the SIP processes rather than model perturbations (Mittermaier, 2007; Miltenberger et al., 2021). The ensemble mean and standard deviation were then calculated for each case. Observational data from field campaigns are used to evaluate and validate the model outputs.

## 3 Results

### 3.1 Model evaluation: storm evolution and radiation

To evaluate the model performance in simulating the Hector thunderstorm, Figure 3 compares visible satellite imagery from MTSAT and simulated outgoing longwave radiation (OLR) at different times during the storm's evolution. In the initial stage, shallow convective clouds are triggered along the sea-breeze fronts over the Tiwi Islands, consistent with the Hector initiation described by Connolly et al. (2013). As convection intensifies, multiple convective cells develop and begin to merge, resulting in the rapid growth of the storm and the formation of a deep convective core (Figures 3a, 3d, and 3g). The model reproduces the organization of convective cells and captures the timing and location of the primary updrafts, as indicated by the observed cloud fields.

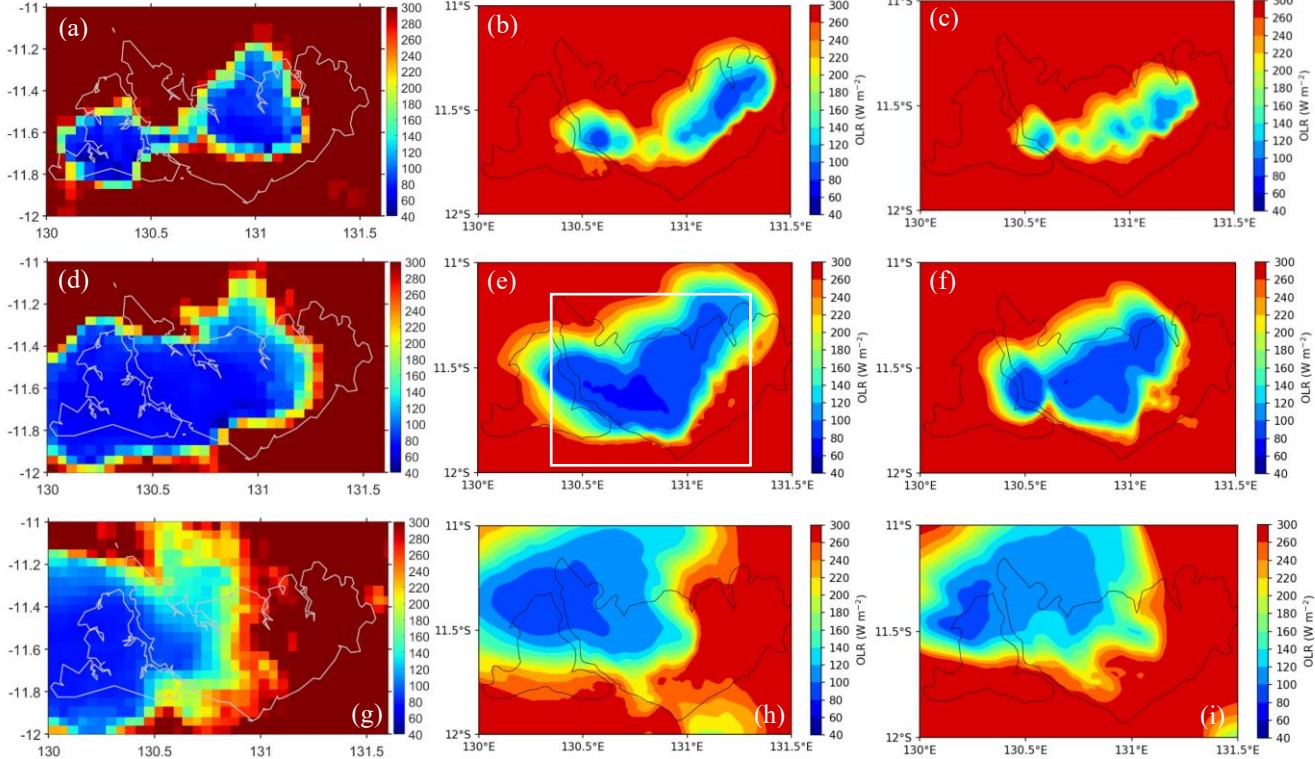

**Figure 3.** MTSAT visible channel images at (a) 05:30 UTC, (d) 06:30 UTC and (g) 08:30 UTC, and simulated outgoing longwave radiation (OLR, unit: W m$^{-2}$) of the (b, e, h) all-SIP and (c, f, i) no-SIP experiments at 04:30 UTC, 05:30 UTC, and 07:30 UTC on 01 December 2005. The simulated convection was earlier than the observation by about 1 h. All panels share an identical color scale. The rectangle in panel (e) denotes the region where the results are analyzed.

During the mature stage, the storm exhibits a well-developed anvil outflow, with the all-SIP simulation capturing the spatial coverage of deep convection more realistically (Figures 3d, e). This is reflected in the broader area of cold cloud tops and extensive low-OLR regions. In contrast, even when the no-SIP experiment produces its most organized cloud system, it remains smaller and less persistent

compared to the all-SIP case, with reduced anvil coverage and duration (Figure 3f). As the storm decays, both simulations show gradual reduced cloud coverage. However, the all-SIP simulation maintains a more extensive anvil cloud during the late stage, in better agreement with observations. Overall, these results demonstrate that including SIP processes improves the model ability to capture the initiation, intensification, and full life cycle of the Hector storm, especially the spatial and temporal evolution of its

convective and anvil cloud components.

We further compared the Contoured Frequency by Altitude Diagram (CFAD) of radar reflectivity derived from CPOL radar observations and simulations, as shown in Figure 4. CFAD represents a time–height plot of the fraction of a defined domain that has a radar reflectivity exceeding a certain threshold (5 dBZ in this paper; Connolly et al., 2013). The domain used to calculate the statistical coverage is shown

in Figure 1. The observed CFAD (Figure 4a) is characterized by a distinct convective core below ~8 km and a broad distribution in upper levels, which is associated with the extensive anvil outflow typical of Hector storms. This anvil region indicates a large amount of ice particles spreading horizontally between 10 and 16 km.

The all-SIP simulation (Figure 4c) captures both the vertical extent and horizontal spread of the anvil, showing a broader and more persistent layer of weak reflectivity aloft, consistent with observations. This indicates that the inclusion of multiple SIP processes enhances the generation and lofting of small ice crystals, which are lifted to the higher levels and contribute to the development of an extensive anvil. In contrast, the no-SIP simulation (Figure 4d) exhibits a much weaker anvil layer, with lower frequencies of reflectivity above 10 km, suggesting a limited ice production and upward ice transport. In this case, SIP yields a higher cloud top and closer agreement with observations, consistent with previous simulations (e.g., Qu et al., 2022). Cloud top responses can vary across configurations. For example, an idealized single cloud study reported lower tops when SIP produced small ice that depleted cloud droplets and limited upper-level ice formation (Grzegorczyk et al., 2025b).

The comparison of vertical profiles of average radar reflectivity is shown in Figure 4b. It demonstrates that the inclusion of SIP leads to a reduction in mean reflectivity values in middle levels, which can be explained by the presence of more numerous small ice-phase particles aloft. Ice particles formed by primary freezing are expected to grow at comparable rates when ice-ice collisional breakup is inefficient, and the additional SIP-generated ice particles shift the overall size distribution toward smaller diameters, leading to weaker reflectivity. In our simulations, the all-SIP experiment exhibits averaged values that are ~2 dBZ lower than those in the no-SIP one between 5 and 12 km. The lower reflectivity in the all-SIP case, together with the extensive anvil region above 10 km, highlights the role of SIP in controlling the microphysical properties and spatial extent of the anvil cloud, which is consistent with findings in previous studies (Connolly et al., 2013; Qu et al., 2022). Below 3 km, the model underestimates the frequency of reflectivity values exceeding 5 dBZ (Figures 4c, d). This likely reflects uncertainties in the representation of warm-rain processes, including cloud-to-rain autoconversion, accretion of cloud water by rain, rain evaporation, and the assumed particle size distributions. Differences between the model reflectivity diagnostic and the observational product near the melting layer could also contribute to the underestimation. The bias is similar in the SIP and no-SIP experiments and therefore does not affect the SIP-related conclusions. Nonetheless, the simulated precipitation totals later in the storm evolution are broadly consistent with observations.

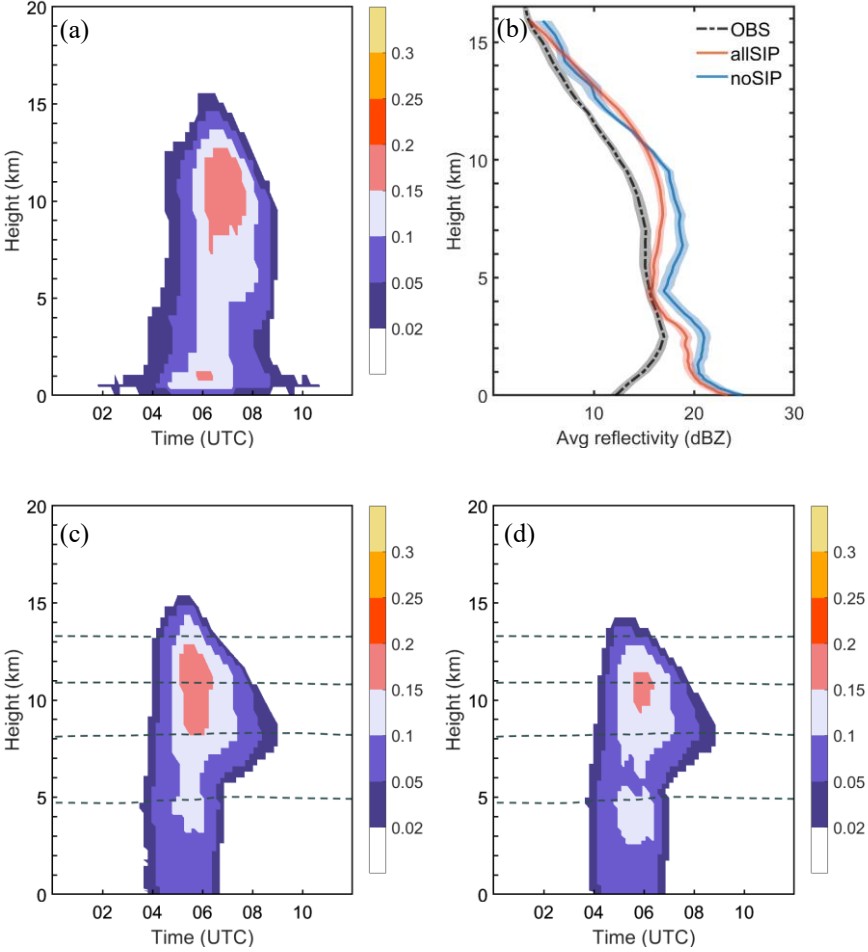

**Figure 4.** Comparisons between the observation and simulations for radar reflectivity. Panels (a), (c), and (d) show the time-height plots of CFADs (fraction of radar reflectivity >5 dBZ): derived from observation, all-SIP, and no-SIP simulations. Panel (b) shows the vertical profiles of averaged radar reflectivity during 05:00–07:00 UTC (04:00–06:00 UTC for simulations), 01 Dec 2005. Shaded areas in (b) show the standard error of each parameter. The 0, -20, -40, and -60°C isotherms are shown by the dashed lines in (c) and (d).

The impact of SIP on the top-of-atmosphere radiative fluxes is shown in Figure 5, which includes the time series of domain-averaged OLR and OSR for various simulations. The inclusion of SIP processes leads to a marked reduction in OLR during the mature stage of convection (Figure 5a), corresponding to the development of extensive and optically thick anvil clouds. The enhanced ice particle number concentration generated by SIP produces more persistent and horizontally widespread anvil clouds, which efficiently reduces outgoing longwave radiation and results in colder OLR signals. For the analysis region shown in Figure 1 during 02:30–08:30 UTC on 01 Dec 2005, the mean OLR in the all-SIP simulation is reduced to 236.2 W m$^{-2}$, compared to 239.4 W m$^{-2}$ in the no-SIP case, indicating a decrease of 1.3%. Similarly, the minimum OLR is 198.4 W m$^{-2}$ in the all-SIP simulation and 202.9 W m$^{-2}$ in the no-SIP case, as illustrated in Table 2.

Meanwhile, SIP also has a significant influence on OSR (Figure 5f). As highlighted in Finney et al.

(2025), the cloud albedo and thus OSR, are highly sensitive to microphysical properties such as ice particle number and size. The increase in small ice particle number concentration within the anvil leads to greater cloud optical thickness and higher shortwave reflectivity. Accordingly, the all-SIP simulation yields a mean OSR of 250.9 W m$^{-2}$, higher than the no-SIP value of 246.5 W m$^{-2}$, representing an increase of 1.8%. The peak OSR also rises, from 339.5 W m$^{-2}$ in the no-SIP case to 353.6 W m$^{-2}$ in the all-SIP simulation. It should be noted that these changes in radiation could arise from a combination of factors, such as variations in cloud area or cloud top temperature. Nonetheless, the overall changes in OLR and OSR remain consistent across SIP simulations, suggesting a substantial influence of SIP on the radiative characteristics of tropical convective systems.

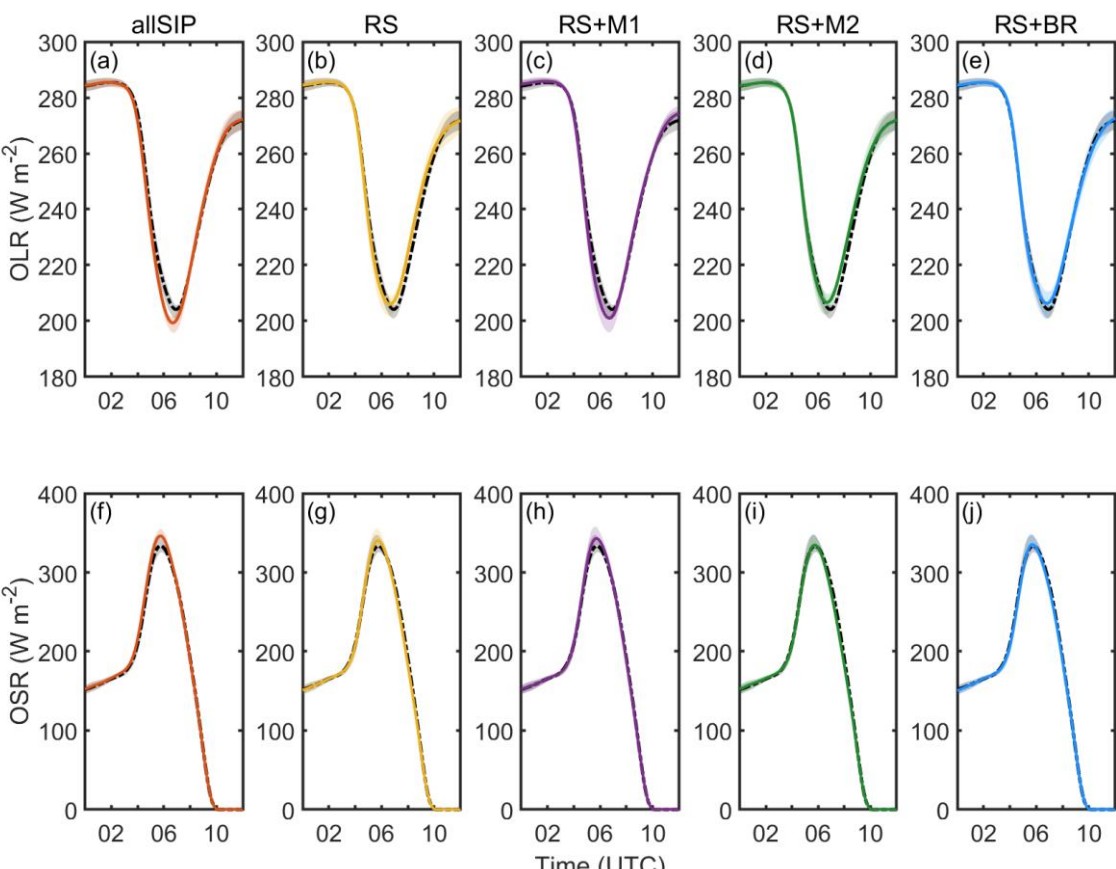

**Figure 5.** Time evolutions of averaged (a-e) outgoing longwave radiation (OLR, unit: W m$^{-2}$) and (f-j) outgoing shortwave radiation (OSR, unit: W m$^{-2}$) for noSIP (dash-dotted) and each SIP configuration (colored), on 01 Dec 2005. The solid lines denote the ensemble means, and the shaded envelopes show the standard error of each parameter. All values are calculated over the 4-member ensemble.

**3.2 Impacts of SIP on precipitation**

Figure 6 presents the spatial distribution of accumulated precipitation during the simulation period, alongside radar-based observations. The observed precipitation field is characterized by an intense convective core over the Tiwi Islands, with accumulated rainfall exceeding 60 mm, surrounded by weaker

stratiform precipitation extending northeastward. Both simulations reproduce the main features of this structure to varying degrees. The no-SIP simulation produces a broader and more diffuse precipitation field, with rainfall spread over a wider area (Figure 6c). For spatial diagnostics, we define a heavy-rainfall region as the area with accumulated rainfall ≥40 mm (black contours in Figures 6b, c; Grzegorczyk et al., 2025b). This heavy-rainfall region is more spread out in the no-SIP than in the all-SIP case. In contrast, the all-SIP simulation shows a more localized and compact precipitation field that better matches the observed spatial pattern (Figure 6b), although both simulations underestimate the observed surface precipitation amount and areal coverage. Several aspects of the configuration may contribute, including km-scale grid spacing (1.5 km), parameter choices in the bulk microphysics scheme, and uncertainties in the environmental forcing. On the observational side, CPOL-based precipitation data also carry documented uncertainties (see Section 2.3). Nevertheless, our analysis focuses on the contrasts between SIP and no-SIP runs, which remain robust across the sensitivity tests. Comparing the two simulations, the no-SIP case yields a slightly higher domain-averaged accumulated precipitation than the all-SIP one (Figure 6d). Relative to no-SIP, all-SIP run encloses a smaller area within the 40 mm contour and features fewer, more compact heavy-rainfall regions (Figures 6b, c), indicating a more localized precipitation pattern.

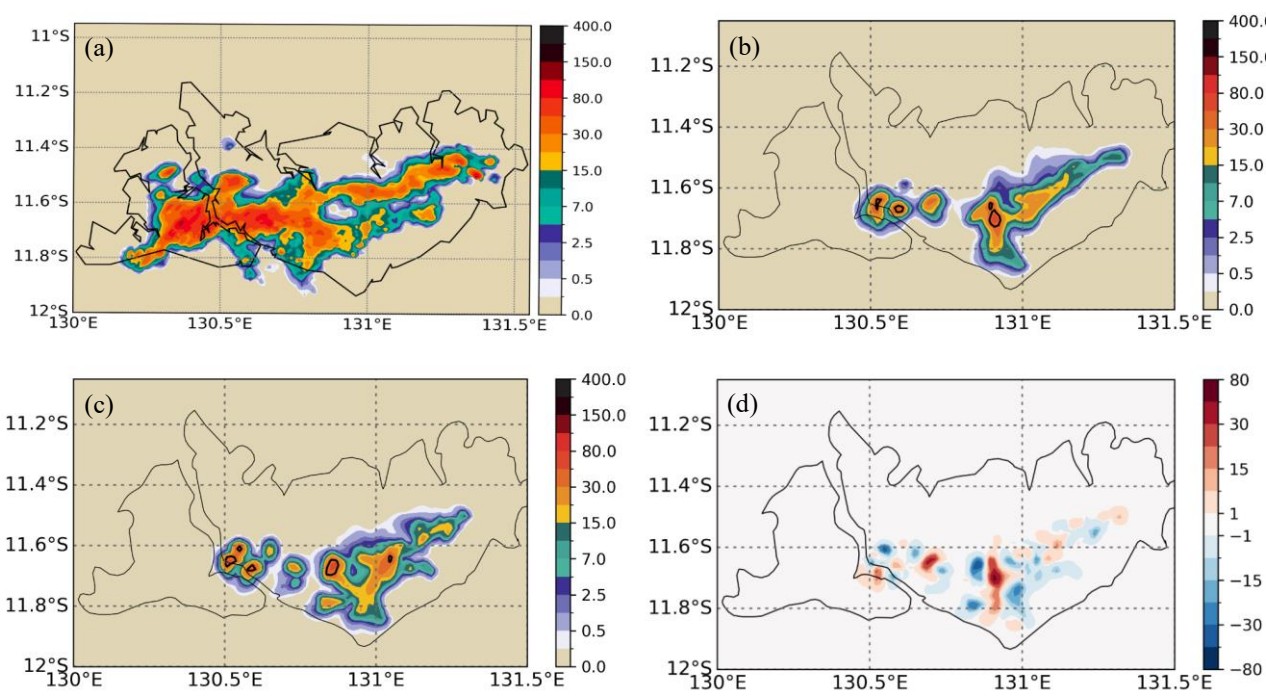

**Figure 6.** Panels (a)–(c) show the spatial distribution of accumulated precipitation (unit: mm) in the (a) observation, (b) all-SIP, and (c) no-SIP simulations during 03:30–09:30 UTC (02:30–08:30 UTC for simulations). Black contours in panels (b)–(c) show areas with accumulated rainfall ≥40 mm. Panel (d) shows the difference (all-SIP minus no-SIP) in

accumulated precipitation between the simulations.

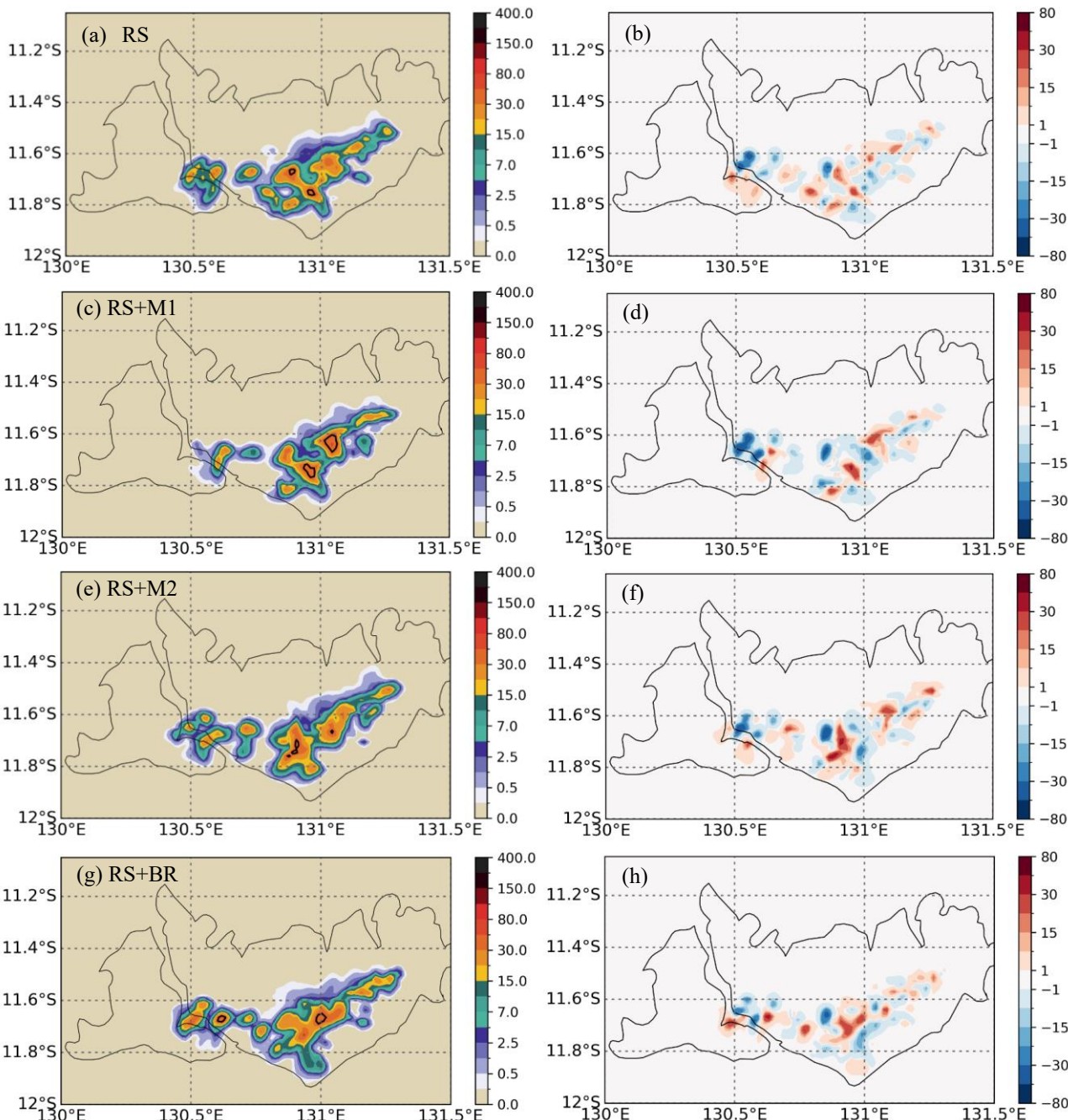

**Figure 7.** The left column shows the spatial distribution of accumulated precipitation (unit: mm) during 02:30–08:30 UTC in the simulations: (a) RS, (c) RS+M1, (e) RS+M2, and (g) RS+BR. Black contours show areas with accumulated rainfall ≥40 mm. The right column shows the deviations of accumulated precipitation (unit: mm) between four of the simulations with the SIP parameterizations in place, and the no-SIP simulation.

Figure 7 presents the spatial distribution of accumulated precipitation for the different SIP combinations and the corresponding deviations from the no-SIP simulation. The RS simulation produces a precipitation pattern broadly similar to the no-SIP case, with relatively small deviations (Figures 7a, b).

Adding the Mode 1 process (RS+M1) yields increases within and adjacent to the convective core, including the northeastern band (Figure 7c, d). The ≥40 mm contour indicates a more compact heavy-rainfall region than in the RS case. The RS+M2 experiment produces localized enhancement that are more fragmented and slightly displaced downwind compared with RS+M1 (Figures 7e, f). The heavy-rainfall region (≥40 mm) appears as separate areas along the band. Including ice–ice collisional breakup (RS+BR) forms a narrow zone of increased heavy precipitation without obvious suppression in the surroundings (Figures 7g, h). Overall, the deviation maps show that individual SIP pathways modify rainfall differently: rime splintering alone has a limited impact, whereas adding Mode 1, Mode 2, or collisional breakup introduces stronger localized increases and primarily reorganizes the spatial distribution.

The domain-averaged precipitation time series in Figure 8a complements the spatial distributions shown in Figure 7, providing additional context for the impact of different SIP mechanisms on total rainfall. During 03:00–07:00 UTC, RS+M2 and RS+BR maintain higher domain-averaged rainfall, yielding larger increases in total precipitation (Figure 8a). The RS+BR simulation shows the highest total amount, suggesting that ice–ice collisional breakup has a comparatively strong effect on overall rain production among the SIP mechanisms tested. The all-SIP simulation produces the lowest domain-averaged precipitation among all experiments, as shown in Figure 8a. Its time-averaged precipitation rate is 0.63 mm h$^{-1}$, approximately 8% lower than the 0.69 mm h$^{-1}$ in the no-SIP case over the same period (03:00–07:00 UTC). This result underscores the non-linear interactions among SIP pathways in deep convection. When multiple SIP processes act simultaneously, the resulting increase in upper-level ice particle concentration may suppress warm-rain production by diverting condensate into small ice particles that are less likely to precipitate efficiently. Additionally, enhanced ice aloft can inhibit the growth of larger rimed hydrometeors and reduce the mass flux toward the surface (Dedekind et al., 2021; Grzegorczyk et al., 2025c). Together, these factors help explain why the all-SIP simulation yields less total rainfall than cases with only one or two active SIP mechanisms. This is consistent with previous studies that emphasized the complex and sometimes competing roles of SIP in cloud microphysics and precipitation efficiency (e.g., Han et al. 2024; Grzegorczyk et al. 2025b, c).

Figure 8b shows the time series of the domain-maximum precipitation rate, representing the evolution of localized extreme rainfall events. All simulations, including no-SIP and those with various SIP mechanisms, produce similar peak magnitudes and timing. The peak rate reaches 199.6 mm h$^{-1}$ in the no-SIP and 186.6 mm h$^{-1}$ in the all-SIP case. The RS+M2 and RS+BR experiments produce slightly higher maxima near 04:00 UTC, but the differences among all cases are relatively small and within

modelling variability. In this case, SIP has limited impact on the domain-maximum precipitation rates. While certain SIP processes slightly enhance localized intensities, their effect on the most extreme point-scale rainfall events remains modest. This contrasts with the more substantial differences seen in accumulated precipitation fields and deviation maps (Figure 7, right column), suggesting that SIP more strongly modifies spatial distribution and total precipitation amount rather than intensifying single-point extremes. It is worth noting that previous studies have reported varying impacts of SIP on precipitation intensity. For example, Grzegorczyk et al. (2025b, c) found that SIP can reduce heavy rainfall (>40 mm), whereas Sullivan et al. (2018) reported localized enhancement of convective regions.

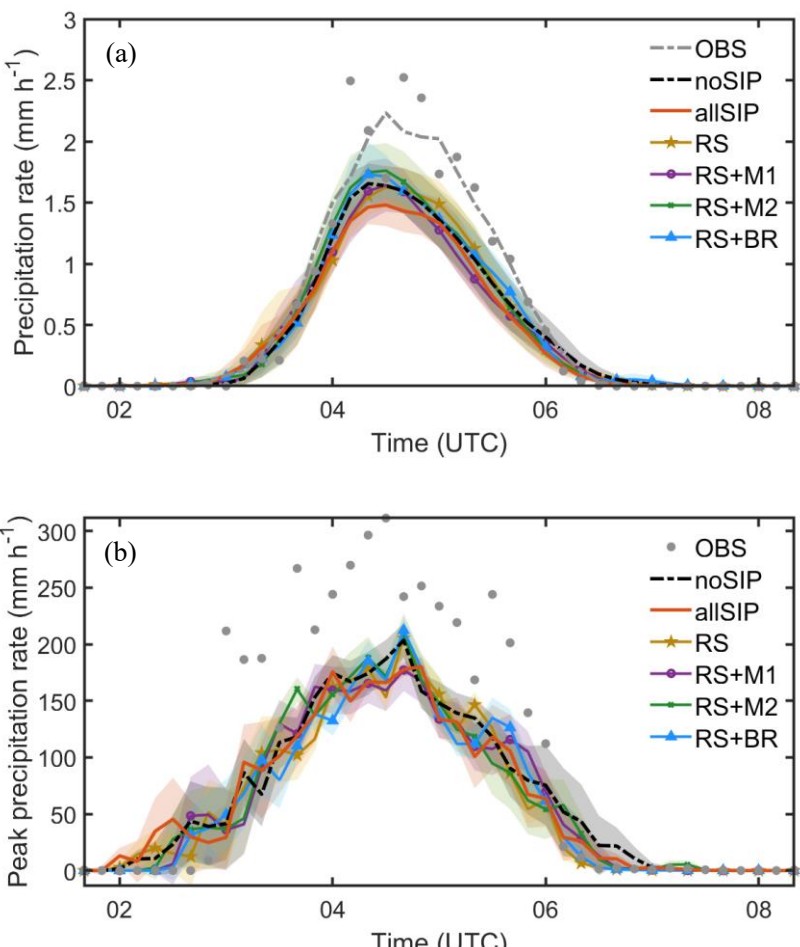

**Figure 8.** Time evolution of (a) domain-averaged precipitation rate, and (b) domain-maximum precipitation rate (unit: mm h$^{-1}$) from CPOL observations and model simulations. For simulations, the solid line represents the ensemble mean, and the shaded area indicates the standard error of each parameter across the 4-member ensemble. The gray dashed line in panel (a) shows a fitted curve to the observations. To facilitate comparison, the time axis for the observations has been shifted by 1 h to align with the simulation results.

### 3.3 Impacts of SIP on microphysical properties

### 3.3.1 Ice number concentration

Figure 9a shows the vertical profiles of horizontally averaged ice crystal number concentration ($N_{ice}$) from various sensitivity experiments. The averages are calculated over grid boxes within the analysis region (Figure 1) where IWC > 0.01 g m$^{-3}$. The all-SIP simulation exhibits peak values of $N_{ice}$ at approximately 8 km, where the temperature T > -20 °C. At T = -5 °C (around 6 km), $N_{ice}$ reaches ~1.5 L$^{-1}$, compared to 0.5 × 10$^{-3}$ L$^{-1}$ in the no-SIP case over regions with IWC > 0.01 g m$^{-3}$, representing an increase of more than 3 orders of magnitude. This enhancement reflects the cumulative contribution of multiple SIP mechanisms in generating secondary ice particles aloft.

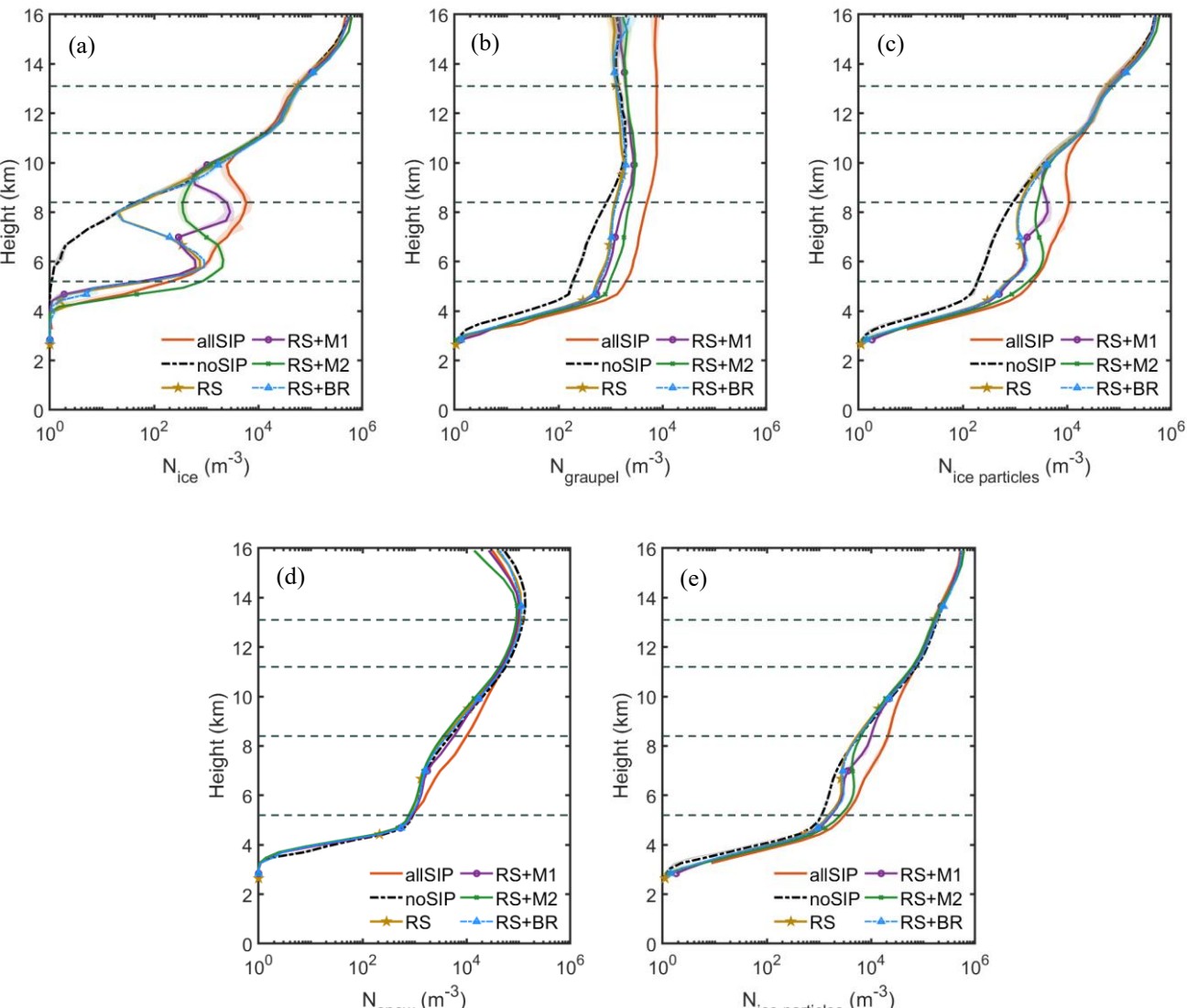

**Figure 9.** Vertical profiles of horizontally averaged number concentration of (a) ice crystals ($N_{ice}$), (b) graupel ($N_{graupel}$), (c) ice-phase particles ($N_{ice\ particles}$; sum of ice crystals and graupel), (d) snow ($N_{snow}$), and (e) total ice-phase particles ($N_{ice\ particles}$; sum of ice crystals, graupel, and snow) from the all-SIP, no-SIP, RS, RS+M1, RS+M2, and RS+BR experiments. All values are averaged over regions with IWC > 0.01 g m$^{-3}$ during 02:30–08:30 UTC on 01 December

2005. The solid line represents the ensemble mean, and the shaded area indicates the standard error of each parameter across the 4-member ensemble. Dashed lines indicate the 0, -20, -40, and -60°C isotherms.

Among the individual SIP experiments, RS+M1 simulation (which includes rime splintering and Mode 1 droplet fragmentation) shows an increase in $N_{ice}$ between 8 and 10 km, reaching ~49% of the all-SIP values. At T ≈ -20 °C, the RS+M1 simulation exceeds the no-SIP case by ~2 orders of magnitude, with peak enhancements of ~150 times. The RS+M2 simulation (rime splintering and Mode 2 droplet-ice collisions) also increases $N_{ice}$ relative to no-SIP, with the enhancement mainly between 6 and 8 km (around T = -5 °C), consistent with the expected activation of Mode 2 in lower layers where supercooled raindrops are more abundant. Its peak at 6 km exceeds 2 $L^{-1}$, more than three orders of magnitude above the no-SIP case at the same level. By contrast, the RS+BR experiment (including ice–ice collisional breakup) differs little from the RS-only case across the column: the profiles are nearly identical and fall within the ensemble spread (Figures 9a, e). This indicates a limited contribution from breakup under the present conditions. A further contribution may arise once concentrations are elevated by other SIP processes. As shown in Figure 9b, graupel number concentrations below 8 km (~1–10 $L^{-1}$) fall within the spread reported by convection-permitting studies (e.g., Phillips et al., 2017b), but are on the high side of aircraft-based observations for tropical convection (e.g., Lasher-Trapp et al., 2016). Higher graupel number generally implies smaller mean size and reduced fall-speed differences, which lowers the graupel-involved collisional kinetic energy and likely explains the limited breakup in this case (Phillips et al., 2017b).

Figure 9c further illustrates the combined number concentration of ice crystals and graupel. The SIP-induced increases in ice number remain evident at T ≈ -20 °C, with all-SIP and RS+M1 cases showing clear separation from the no-SIP simulation. For example, at ~8.5 km, the all-SIP reaches a peak value approximately 20 $L^{-1}$, while RS+M1 yields about 2.5 $L^{-1}$, both higher than the no-SIP case (~0.5 $L^{-1}$) in regions with IWC > 0.01 g m$^{-3}$. This indicates that SIP processes not only enhance ice particle production but also help maintain elevated ice amounts at upper levels (see Supplementary Figure S1), consistent with the extensive anvil clouds in Figure 3.

In contrast, snow number concentration (Figure 9d) exhibits limited sensitivity to SIP processes. Snow is produced mainly between 8 and 12 km, and the differences among experiments are minimal, indicating that snow production is dominated by aggregation rather than SIP-induced ice multiplication. Because snow varies little across experiments, snow-involved breakup is likely similar in magnitude across runs and cannot explain the contrasts in ice number. Under warm profiles, graupel-involved

collisions exhibit limited energetics, with smaller mean sizes and weaker mass-weighted fall-speed contrasts, which reduces the collisional kinetic energy required for efficient breakup (Phillips et al., 2017b). Consistent with this, the simulation with breakup enabled (RS+BR) shows ice-number profiles similar to the RS-only run (Figures 9a, e). Figure 9e summarizes the total ice-phase particle number concentration (sum of ice crystals, graupel, and snow). Although including snow reduces the relative difference between SIP and no-SIP experiments, the all-SIP case ($\sim$20 L$^{-1}$) remains up to five times higher than the no-SIP simulation ($\sim$4 L$^{-1}$) at 8 km, owing to sustained enhancements in ice and graupel. Overall, under the present convective conditions, SIP processes, particularly those involving drop fragmentation (Mode 1 and Mode 2), play the dominant role in increasing ice crystal number, whereas collisional breakup contributes little. These changes in ice particle populations modify the cloud microphysics and can influence cloud dynamics and radiative properties. Consistent with this, the reflectivity decrease in Figure 4 is mainly associated with the increase in small ice crystal number; graupel contributes minor changes relative to crystals, whereas snow remains weakly sensitive to SIP.

### 3.3.2 Ice water content and updraft velocity

Figure 10 presents the vertical profiles of horizontally averaged ice water content (IWC) and maximum updraft velocity from the different SIP sensitivity experiments.

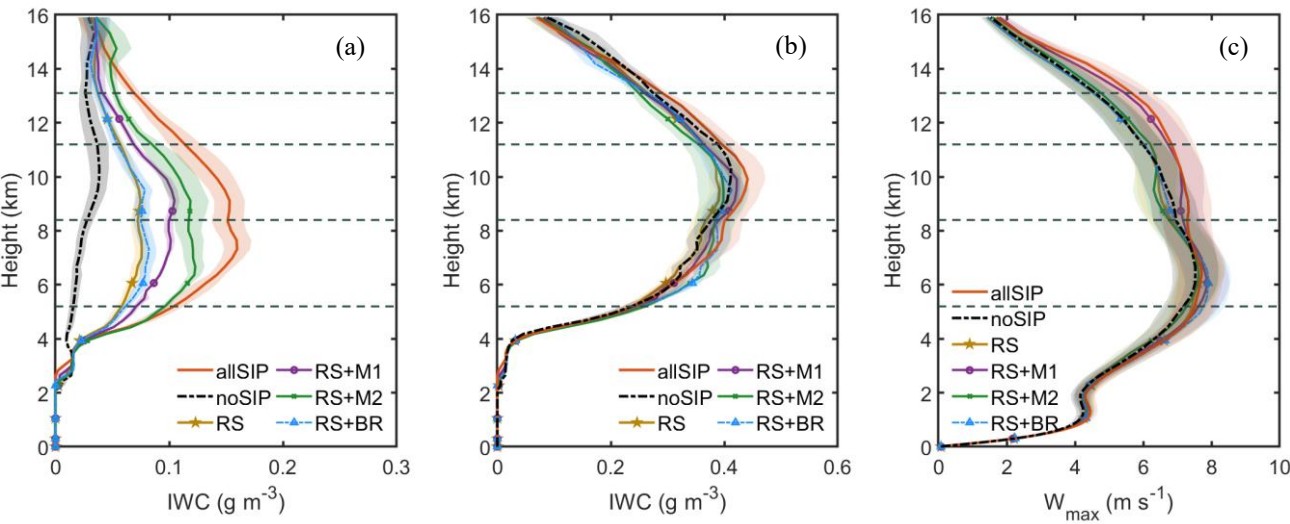

**Figure 10.** Vertical profiles of (a) ice water content (IWC) from the combinations of ice crystals and graupel, (b) IWC including ice crystals, graupel, and snow, and (c) maximum updraft velocity ($W_{max}$). Profiles are calculated from 02:30 to 08:30 UTC, 01 December 2005. Panels (a) and (b) show horizontal averages calculated over regions with IWC > 0.01 g m$^{-3}$, and panel (c) depicts the domain-wide horizontal maximum at each height level. Solid lines represent the ensemble means, with the shaded areas indicate the standard error across the 4-member ensemble. The dashed lines represent isotherms of 0, -20, -40, and -60°C.

Figure 10a shows IWC including ice crystals and graupel. The all-SIP simulation exhibits the highest IWC values at ~8 km (T > -20 °C), with a clear enhancement compared to the no-SIP case. This enhancement explains the improved representation of radar reflectivity compared to observations, particularly above 6 km (Figure 4). The IWC in the all-SIP run is approximately 0.16 g m$^{-3}$, compared to 0.03 g m$^{-3}$ in the no-SIP case, reflecting a more than 400% increase. Among the individual SIP experiments, RS+M1 shows IWC levels similar to all-SIP between 8 and 11 km, indicating the strong role of Mode 1 droplet fragmentation in enhancing upper-level ice mass. At 9 km, the IWC in RS+M1 reaches ~0.10 g m$^{-3}$, over three times higher than the no-SIP value. The RS+M2 case also increases IWC relative to no-SIP but with a more confined impact between 6 and 10 km. Its peak IWC near 7 km (~0.12 g m$^{-3}$) lies between RS+M1 and all-SIP, suggesting that Mode 2 is particularly effective in the lower region. In contrast, RS+BR and RS-only experiments produce nearly overlapping IWC profiles throughout the column, with values at 7 km remaining below 0.06 g m$^{-3}$, suggesting that ice–ice collisional breakup remains inactive under the present conditions.

When snow is included (Figure 10b), column IWC increases across all simulations, with the largest enhancements above ~9 km (T ≈ -20 °C) where snow dominates the mass budget. In this layer, snow is both more numerous than graupel (Figures 9b, d) and larger in size than ice crystals, so it contributes a greater share of IWC. Once snow is accounted for, the inter-experiment differences that mainly arise from crystals are partly diluted (Figures 10a, b), as graupel remains limited. Even so, SIP still yields a net IWC increase between 8 and 12 km. At 10 km, all-SIP reaches 0.43 g m$^{-3}$ compared to 0.39 g m$^{-3}$ in no-SIP (~+10%), with differences of 5–15% across 8–12 km despite the mitigating effect of snow.

These patterns are consistent with the number concentration profiles (Figures 9d, e). SIP substantially increases ice crystal numbers (and to a lesser extent graupel) in the upper troposphere and anvil, whereas snow shows weak sensitivity to SIP. Consequently, adding snow reduces the contrast among experiments, yet the SIP-driven shift toward more numerous small crystals remains evident, explaining the decrease in mean reflectivity aloft (Figure 4b).

The differences in IWC are accompanied by variations in updraft velocity, as shown in Figure 10c. Figure 10c presents the vertical profiles of the domain-maximum updraft velocity ($W_{max}$). Above 9 km, all SIP experiments have $W_{max}$ ~10% higher than the no-SIP simulation, with the largest enhancements in the all-SIP and RS+M1 cases, particularly between 10–12 km. This enhancement can be attributed to the increased ice crystal number concentrations induced by SIP processes (Figure 9a), which promote more efficient depositional growth and freezing of supercooled water in the upper layers. These processes

enhance latent heat release, thereby reinforcing local buoyancy and supporting stronger updrafts (Qu et al., 2022; Grzegorczyk et al., 2025b). Below 8 km, the differences among experiments are minimal, which is likely because the low-level updrafts are primarily driven by dynamical processes such as boundary-layer convergence and cold pool outflows, and are therefore less sensitive to microphysical modifications induced by SIP. Given the km-scale grid (1.5 km) used here, the peak vertical velocities are smaller than in higher-resolution large-eddy simulations (e.g., ~100 m; Dauhut et al., 2015). This likely reflects resolution effects that weaken intense updrafts, for example through numerical diffusion and subgrid-scale treatment. The contrasts between SIP and no-SIP simulations remain robust, as demonstrated by the sensitivity experiments.

Overall, these results demonstrate that SIP processes, particularly those involving droplet fragmentation (Mode 1 and Mode 2), substantially enhance ice mass and ice crystal number concentration in the upper troposphere and anvil region. While the inclusion of snow reduces the relative difference between SIP and no-SIP cases, the enhancement of ice-phase particles by SIP remains evident, highlighting its important role in modifying cloud microphysical structure in the convective and anvil regions.

## 4 Discussion

Based on model simulations, our results demonstrate that SIP processes significantly enhance both the ice particle population and IWC in the upper cloud layers (Figures 9 and 10). For instance, at -5 °C (~6 km), the ice crystal number concentration increases from $0.5 \times 10^{-3}$ L$^{-1}$ in the no-SIP case to 1.5 L$^{-1}$ in the all-SIP simulation, representing an enhancement of over 3 orders of magnitude. At ~8.5 km, all-SIP and RS+M1 cases reach peak values of 11.5 L$^{-1}$ and 2.5 L$^{-1}$, compared to 0.86 L$^{-1}$ in the no-SIP case. The RS+M2 increases ice number concentrations primarily between 6 and 8 km (T = -5 °C), consistent with Mode 2 activation in regions rich in supercooled raindrops. For the IWC, SIP processes increase values at 8–9 km from 0.03 g m$^{-3}$ in the no-SIP case to 0.10 g m$^{-3}$ in RS+M1, and 0.16 g m$^{-3}$ in the all-SIP simulation, representing an enhancement of over 400%. These results highlight the dominant role of SIP, especially Mode 1 and Mode 2 droplet fragmentation, in generating high concentrations of small ice particles and increased IWC in the upper troposphere.

These microphysical changes lead to modifications in radiative fluxes. The reduced OLR is mainly attributed to the colder and more optically thick cloud tops. With higher concentration of small ice particles (Figure 4b), OSR is enhanced due to the increased shortwave scattering and cloud reflectivity.

The mean OLR decreases from 239.4 W m$^{-2}$ in the no-SIP case to 236.2 W m$^{-2}$ in the all-SIP simulation, with a minimum as low as 198.4 W m$^{-2}$. Similarly, OSR increases from a mean of 246.5 W m$^{-2}$ in no-SIP to 250.9 W m$^{-2}$ in all-SIP, with a peak value of 353.6 W m$^{-2}$. As emphasized by Finney et al. (2025), even a 1–3% increase in high cloud albedo can produce substantial radiative feedback. While we do not explicitly calculate cloud albedo, the observed changes in longwave and shortwave reflectivity suggest that SIP could play an important role in modulating the radiative forcing of convective systems, potentially influencing regional and large-scale climate feedback. Further work evaluating these effects across different events and environments, and using long integrations with parameterized SIP, would be helpful for quantifying their climate relevance.

SIP processes, particularly those involving droplet fragmentation (Mode 1 and Mode 2), also affect precipitation structure in the Hector storm. By increasing ice crystal concentrations and IWC in the upper levels, SIP promotes more localized, compact heavy-rainfall region near the main convective zone (Figure 6). However, the domain-averaged precipitation rate is reduced (Figure 8a), with the all-SIP case producing 0.63 mm h$^{-1}$, about 8% lower than the 0.69 mm h$^{-1}$ in no-SIP simulation. These results are similar to the findings from Han et al. (2024) and Grzegorczyk et al. (2025b, c), suggesting that in the mixed-phase region and above, SIP shifts more condensate from warm-rain processes toward ice-phase pathways, producing less surface precipitation. Additional losses may arise from sublimation of small ice particles before reaching the ground. In the RS case (rime-splintering only), changes to surface rainfall are small, consistent with limited graupel growth and the narrow temperature window for HM activation. Adding Mode 1 produces numerous small ice particles at relatively higher levels ($\sim$−15 to −25 °C; Figure 9a), which likely shifts condensate from warm-rain processes and places much of the newly formed ice farther from the melting layer. Aggregation and/or riming therefore tend to occur later, and sublimation during descent can limit increase in surface rainfall. Mode 2 generates secondary ice closer to the melting level ($\sim$−5 °C), often within regions of higher liquid water content, which can accelerate riming to snow/graupel. The shorter distance to the surface can reduce sublimation losses and raise the domain-mean precipitation. Including ice–ice collisional breakup (BR) produces additional fragments within the mixed-phase layer that are likely to be incorporated into ongoing riming, reinforcing a narrow band of enhancement near the convective region and yielding increased surface rainfall, despite $N_{ice}$ remaining close to RS. When multiple SIP processes act together (all-SIP), the increase in small ice aloft tends to reduce the liquid water available for warm-rain collection and riming, helping to explain why total rainfall can be lower than in cases with only one or two mechanisms, even though precipitation is more localized

(Figures 6 and 8). In contrast, peak rainfall rates change little (Figure 8b): the all-SIP simulation peak is 186.6 mm h$^{-1}$, slightly lower than 199.6 mm h$^{-1}$ in the no-SIP, and all simulations underestimate the observed peak (> 250 mm h$^{-1}$). This indicates that the most extreme precipitation events are primarily driven by dynamical forcings, such as boundary-layer convergence and cold pool outflows, rather than microphysical variations alone (Connolly et al., 2013). While SIP may enhance upper-level buoyancy through latent heat release from depositional growth and freezing, the updraft velocity ($W_{max}$) increases by ~10% (Figure 10c), which is measurable but has limited impact on peak rainfall metrics in this case. Previous studies reported varied impacts of SIP on precipitation intensity: reductions in the heavy rainfall (e.g., >40 mm) have been documented (Grzegorczyk et al., 2025b, c), whereas localized enhancements within convective regions have also been reported (Sullivan et al., 2018). The effectiveness of such cold-phase invigoration mechanisms remains uncertain, with some studies suggesting that the added buoyancy may be offset by the increased mass loading from ice particles (Grabowski and Morrison, 2020; Varble et al., 2023). Overall in this case, SIP mainly reshapes the spatial distribution and partitioning/yield of surface rainfall, with limited impact on the intensity of localized extreme events.

To assess the robustness of SIP-induced signals relative to natural meteorological variability, we compared the differences between SIP and no-SIP simulations against the ensemble spread from four members initialized at different times (Figure 11). Overall, the changes caused by SIP in cloud properties (e.g., IWC and ice number concentration) and radiation (OLR) are both systematic and statistically significant, with magnitudes exceeding the ±1σ ensemble spread across all members. For example, while the ensemble spread in domain-averaged OLR is ~1%, the SIP-induced reduction exceeds 1.3%. For IWC and ice number concentration, the differences exceed ensemble variability by more than a factor of two. Other variables, such as OSR and domain-averaged precipitation, show consistent SIP-related responses across ensemble members, but the magnitudes fall closer to the spread, indicating weaker but still coherent signals. In contrast, maximum precipitation rates show little sensitivity to SIP and lie within ensemble spread, suggesting that extremes are highly influenced by meteorological variability. These results suggest that, while natural variability modulates the amplitude of individual responses, the key signals induced by SIP, especially in cloud and radiative properties, are physically robust and statistically significant.

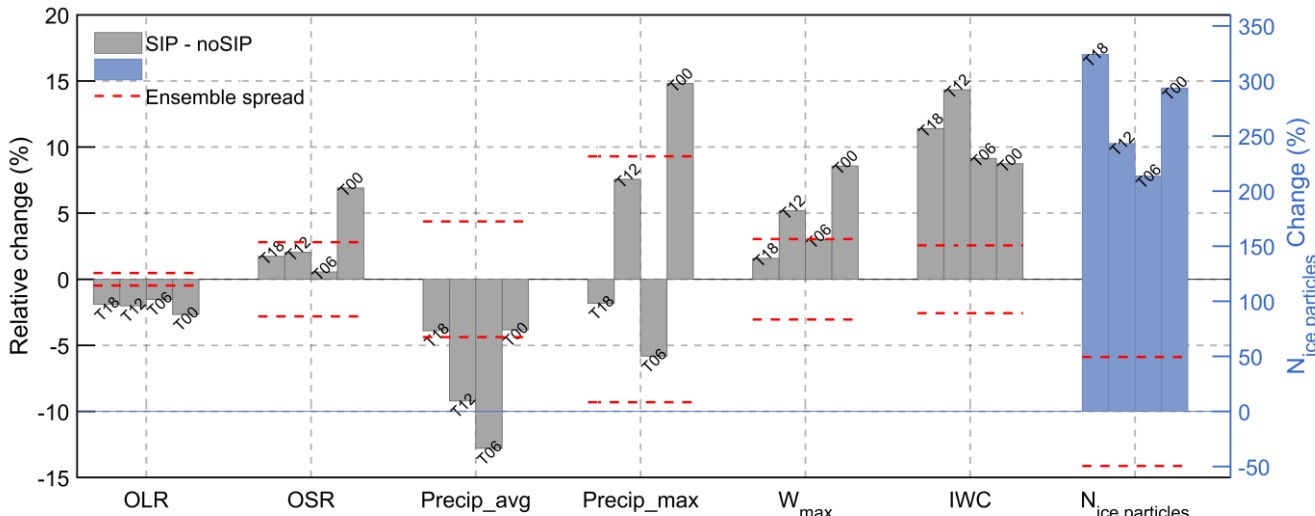

**Figure 11.** Relative change (%) between the SIP-included and no-SIP simulations for different variables: outgoing longwave radiation (OLR), outgoing shortwave radiation (OSR), domain-averaged and -maximum precipitation rate (Precip_avg, Precip_max), domain-maximum updraft velocity ($W_{max}$), ice water content (IWC, including ice crystals, graupel, and snow), and total ice-phase particles number ($N_{ice\ particles}$). Results are shown for four ensemble members initialized at different times (T18, T12, T06, and T00). For each member, the relative change was estimated using 1000 bootstrap samples. Gray and blue bars show the overall relative change for each variable; with $N_{ice\ particles}$ plotted against the right y-axis. Red dashed lines indicate the $\pm 1\sigma$ ensemble spread, computed as the average standard deviation of the bootstrap samples across all members.

Unlike studies that reported strong ice–ice collisional breakup (e.g., Sullivan et al., 2018; Grzegorczyk et al., 2025a), our case shows no clear BR signal (Figure 9). Near ~8 km, graupel number concentrations peak at ~1–10 $L^{-1}$ (Figure 9b). BR remains weak because graupel-involved collision energetics are limited under warm profiles. At comparable IWC, a higher graupel number implies smaller mean size and weaker mass-weighted fall speed contrasts, which reduces the collisional kinetic energy that controls breakup efficiency as formulated by Phillips et al. (2017b). Numerical experiments suggest that graupel–snow interactions provide the major source of breakup fragments under convective conditions (Phillips et al., 2017b). Therefore, the overall BR contribution remains small when graupel-involved collision energetics are limited. These results likely highlight the environmental dependence of ice–ice breakup, which appears less effective in tropical deep convection with warm, moist profiles and limited graupel growth (e.g., Phillips et al., 2017a; Korolev et al., 2020; Huang et al., 2022). In light of this, the existing SIP parameterizations, many of which were developed in mid-latitude cloud environments, should be applied with caution in tropical settings. To improve model performance, future schemes may need to better account for environmental factors, such as graupel abundance, temperature profiles, and convective intensity. Moreover, the interactions between SIP and aerosol conditions remain poorly constrained and deserve further investigation, given their potential influence on cloud

microphysics and storm development in tropical regions (e.g., Sun et al., 2021, 2024).

This study focuses on a single Hector storm case. While the ensemble results provide insights into SIP impacts in tropical deep convection, the findings should be interpreted cautiously before generalized to other storm types or regions. In addition, SIP parameterization remains poorly constrained despite recent advances (e.g., James et al., 2021). More systematic laboratory and field investigations are needed to improve its quantification. Future work should involve multi-case, multi-region simulations and integrate observational constraints to evaluate SIP processes more comprehensively.

**5 Conclusion**

In this study, we first implemented secondary ice production (SIP) parameterizations into the double-moment cloud microphysics scheme of Unified Model (UM-CASIM), including droplet shattering (Mode 1 and Mode 2) and ice–ice collisional breakup. These schemes were applied to a real-case simulation of a Hector tropical deep convective storm to assess their impacts on cloud microphysics, radiative fluxes, precipitation, and storm dynamics. To complement the sensitivity experiments, we conducted a time-lagged ensemble with varied initial times to evaluate the robustness of SIP-induced signals. The changes caused by SIP in cloud properties (e.g., IWC and ice number concentration), radiative fluxes (OLR), and domain-averaged precipitation were generally larger than or comparable to the ensemble spread, indicating that the signals are systematic and not dominated by internal meteorological variability. While OSR showed less consistent significance, the overall patterns support that the microphysical and radiative responses to SIP are physically robust.

Our results demonstrate that SIP, particularly droplet fragmentation processes including Mode 1 and Mode 2, can substantially enhance ice crystal number concentrations and upper-level ice water content (IWC), especially in the anvil region. These microphysical changes modify cloud radiative properties, leading to reduced outgoing longwave radiation (OLR) and increased outgoing shortwave radiation (OSR) at the top of the atmosphere, resulting in better agreement with satellite observations.

SIP also reshapes precipitation characteristics by promoting a more localized, compact rainfall pattern near the main convective zone, while reducing domain-averaged precipitation. This reduction exceeds the ensemble spread and is consistent with a shift from warm-rain to ice-phase growth, accompanied by in-transit losses from sedimentation and sublimation and yielding less surface rainfall. In contrast, peak rainfall rates across the domain remain largely unaffected and fall within ensemble variability, consistent with the minimal changes in maximum updraft velocity ($W_{max}$). These results

715 suggest that SIP primarily modulates the pattern and efficiency of rainfall rather than intensifying extreme convective events.

Among the SIP mechanisms examined, ice-ice collisional breakup had limited influence in this case, likely due to the warm, moist environment and insufficient graupel production. This highlights the environmental dependence of SIP efficiency in tropical convection. These findings suggest that existing 720 parameterizations may need to be adapted when applied to warmer convective conditions. While this study provides new insights into SIP behavior in tropical settings, further works are needed to explore how these processes vary across different storm types and environmental conditions.

## Appendix A

**Table A1. List of symbols.**

| Symbol | Description | Value and units |
|---|---|---|
| $A$ | Number density of the breakable asperities in the contact region | – |
| $C$ | Asperity-fragility coefficient | – |
| $c_w$ | Specific heat capacity of liquid water | 4200 J kg$^{-1}$ K$^{-1}$ |
| $D_1, D_2$ | Diameters of the colliding ice-phase particles in ice–ice collisional breakup | m |
| $DE$ | Dimensionless energy | – |
| $DE_{crit}$ | Critical value of dimensionless energy for onset of splashing | 0.2 |
| $D_i$ | Diameter of ice particles in Mode 1 and Mode 2 | m |
| $D_R$ | Diameters of raindrops in Mode 1 and Mode 2 | m |
| $D_{i,thresh}$ | Diameter of an ice particle whose mass equals that of the colliding raindrop | m |
| $D_{R,thresh}$ | Minimum raindrop diameter for Mode 2 | 0.15 mm |
| $F$ | Interpolation function for the onset of fragmentation | – |
| $f(D_1), f(D_2)$ | Size distribution functions at diameters $D_1, D_2$ | – |
| $f(D_i), f(D_R)$ | Size distribution functions at diameters $D_i, D_R$ | – |
| $f(T)$ | Mass fraction of drop frozen in Mode 2 | – |
| $f_{RS}(T)$ | Temperature-dependent function of rime splintering | – |
| $K_{0(CB)}$ | Collisional kinetic energy at impact | J |
| $L_f$ | Specific latent heat of freezing | $3.3 \times 10^5$ J kg$^{-1}$ |
| $m_1, m_2$ | Mass of colliding ice particles | kg |
| $M_{I0}$ | Mass of each splinter | kg |
| $N_{CB}$ | Number of ice particles due to ice–ice collisional breakup | – |
| $N_{MIL}$ | Total number of large ice particles due to Mode 1 | – |
| $N_{MIT}$ | Total number of ice particles due to Mode 1 | – |

| Symbol | Description | Units |
|---|---|---|
| $P_{gacw}$ | Riming rate of cloud droplets by graupel | number kg$^{-1}$ s$^{-1}$ |
| $P_{ihal}$ | Splinter production rate | number kg$^{-1}$ s$^{-1}$ |
| $P_{sacw}$ | Riming rate of cloud droplets by snow | number kg$^{-1}$ s$^{-1}$ |
| $S_e$ | Surface energy | J |
| $T$ | Freezing temperature of water drop | °C |
| $t$ | Time | s |
| $T_0$ | Value of T at maximum of Lorentzian function for Eq. (5) | °C |
| $T_{B0}$ | Value of T at maximum of Lorentzian function for Eq. (6) | °C |
| $v(D_1), v(D_2)$ | Fall speed of ice particles with diameters $D_1$, $D_2$; denoted $v_1$, $v_2$ for the colliding particles | m s$^{-1}$ |
| $v(D_R), v(D_i)$ | Fall speed of raindrops and ice particles with diameters $D_R$, $D_i$ | m s$^{-1}$ |
| $\alpha$ | Equivalent spherical area of the colliding particle | m$^2$ |
| $\beta$ | Parameter in Eq. (5) | K$^{-1}$ |
| $\beta_B$ | Parameter in Eq. (6) | K$^{-1}$ |
| $\gamma$ | Parameter of riming intensity | – |
| $\gamma_{liq}$ | Surface tension of liquid water | 0.073 J m$^{-2}$ |
| $\zeta$ | Intensity of Lorentzian function in Eq. (5) | – |
| $\zeta_B$ | Intensity of Lorentzian function in Eq. (6) | – |
| $\eta$ | Half-width of Lorentzian function in Eq. (5) | °C |
| $\eta_B$ | Half-width of Lorentzian function in Eq. (6) | °C |
| $\Phi$ | Probability of any drop in Mode 2 containing ice | 0.3 |
| $\Omega$ | Interpolating function for the onset of fragmentation | – |
| $\frac{\partial n_{ice}}{\partial t}\big|_{CB}$ | Production rate of ice particles due to ice–ice collisional breakup | number m$^{-3}$ s$^{-1}$ |
| $\frac{\partial n_{ice}}{\partial t}\big|_{M1,c}$ | Production rate of ice particles from raindrop–ice collisional breakup in Mode 1 | number m$^{-3}$ s$^{-1}$ |
| $\frac{\partial n_{ice}}{\partial t}\big|_{M1,f}$ | Production rate of ice particles from freezing droplet shattering in Mode 1 | number m$^{-3}$ s$^{-1}$ |
| $\frac{\partial n_{ice}}{\partial t}\big|_{M1}$ | Production rate of ice particles due to Mode 1 | number m$^{-3}$ s$^{-1}$ |

$$\frac{\partial n_{ice}}{\partial t}\bigg|_{M2}$$     Production rate of ice particles due to Mode 2     number m$^{-3}$ s$^{-1}$

*Data availability.* The model output in this study is archived on the UK Met Office MASS system (suite ID: u-dp252) and accessible via the JASMIN platform (http://www.jasmin.ac.uk/, last access: 10 Nov 2025). The CASIM microphysics code, including the newly implemented SIP schemes, is available on the Met Office Science Repository Service (MOSRS): https://code.metoffice.gov.uk/trac/monc/log/casim/branches/dev/mengyusun/r10208_SIP_rep (last access: 10 Nov 2025). Specific SIP configurations are stored under revisions: rev11607 (all-SIP), rev11609 (no-SIP), rev11678 (RS), rev11679 (RS+M1), rev11680 (RS+M2), and rev11677 (RS+BR).

*Author contributions.* MS and PJC conceptualized the study and designed the experiments. PJC led the implementation of the SIP parameterizations in CASIM, with assistance from MS. PRF contributed to the UM setup and advised on the CASIM code and ensemble methodology. MS carried out the simulations and formal analysis and wrote the paper, with support from PJC, PRF, DLF, and AMB. AMB supervised the project and acquired funding.

*Competing interests.* The authors declare that they have no conflict of interest.

*Acknowledgements.* This work was supported by the Natural Environment Research Council (NERC) through the DCMEX project (grant NE/T006420/1). This project (author PJC) has received funding from Horizon Europe programme under Grant Agreement No. 101137680. We thank the Met Office and NERC for access to the Monsoon high-performance computing system, and the UK Centre for Environmental Data Analysis (CEDA) for data storage and processing via the JASMIN infrastructure (https://doi.org/10.1109/BigData.2013.6691556).

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

**Tables**

## Table 1. List of experiments

| | Type of secondary ice production | | | |
|---|---|---|---|---|
| | Rime splintering | Droplet shattering (Mode 1) | Droplet shattering (Mode 2) | Ice–ice collisional breakup |
| all-SIP | Hallet and Mossop (1974) | Phillips et al. (2018) | Phillips et al. (2018) | Phillips et al. (2017) |
| RS | Hallet and Mossop (1974) | — | — | — |
| RS+M1 | Hallet and Mossop (1974) | Phillips et al. (2018) | — | — |
| RS+M2 | Hallet and Mossop (1974) | — | Phillips et al. (2018) | — |
| RS+BR | Hallet and Mossop (1974) | — | — | Phillips et al. (2017) |
| no-SIP | — | — | — | — |

Note: Mode 1 (M1) represents fragmentation during spherical drop freezing, and Mode 2 (M2) represents collisions of supercooled raindrops with more massive ice (Phillips et al. 2018).

## Table 2. Comparison of radiation under different SIP configurations

| | Mean OLR (W m$^{-2}$) | Min OLR (W m$^{-2}$) | Mean OSR (W m$^{-2}$) | Max OSR (W m$^{-2}$) |
|---|---|---|---|---|
| all-SIP | 236.20 | 198.39 | 250.95 | 353.58 |
| RS | 239.99 | 205.27 | 245.39 | 346.60 |
| RS+M1 | 237.36 | 199.96 | 248.45 | 348.52 |
| RS+M2 | 240.66 | 205.85 | 243.18 | 341.29 |
| RS+BR | 239.94 | 205.41 | 244.45 | 340.60 |
| no-SIP | 239.36 | 202.92 | 246.47 | 339.54 |

Note: All values are calculated over the analysis region shown in Figure 1, during 02:30–08:30 UTC on  01 December 2005.