# Peer review of "Influence of secondary ice formation on tropical deep convective clouds simulated by the Unified Model"

_EGUsphere, 2025_

## Referee Comment (RC2)

Influence of Secondary Ice Formation on Tropical Deep Convective Clouds
Simulated by Unified Model

The paper numerically investigates the possible role of secondary ice production (SIP) in forming overall ice number concentration, and its influence on cloud properties in a convective case observed during ACTIVE campaign in early December 2005. For this, the authors incorporated three SIP processes (Hallett-Mossop rime-splintering, ice-ice collision, breakup of freezing raindrop) in UK Met Office Unified Model's 2-moment CASIM microphysics scheme. The study finds that, through increased ice number and mass in the upper region, SIP can modify the anvil structure in the simulated thunderstorm, and changes the precipitation formation, especially associated with the convective core. The ensemble simulations are also performed that illustrates the robustness of the presented results. The introduction is comprehensive, and sufficiently discusses the recent advancements and challenges in SIP research, and the methodology is sound. The simulated properties, such as the radar reflectivity, OLR, and precipitation are compared and validated against the observations. While the overall presentation is good, the paper could benefit from a more detailed discussion and comparison of how the simulated microphysical properties agrees well with the observations. Nevertheless, the study addresses a timely and important topic and may be considered for publication after satisfactorily addressing the following concerns.

**General comments**

More details of radiation and microphysics scheme used are needed. The presented validation is reasonable but could have benefited from additional comparison, such as observed liquid/ice properties, if such observations (from satellite or other platforms) are available. Also, many findings presented in the manuscript can be supported by some previous studies, and can also be acknowledged in the introduction section to strengthen the proposed research questions.

**Specific comments**

**Abstract**

Line 14: Since the study mainly quantifies the impacts of SIP on the simulated clouds without attempting to modify/improve the existing SIP parameterizations (beyond the use of revised $\Phi$ from James et al. 2021), I would suggest rephrasing this sentence, as the focus is not on quantifying and reducing uncertainties in the modelled SIP processes.

Line 16: Mode 1 and 2 of drop shattering are scheme-specific terms and are not widely recognized. These are proposed by Phillips et al. (2018) to represent drop shattering in collision between rain/drizzle drop with ice particle. Other schemes of SIP in drop shattering (e.g., Sullivan et al. 2018) only consider shattering of raindrop during freezing, initiated due to immersed INP, without separating mode 1 and 2. Better to omit using mode 1 and mode 2 and say only 'drop fragmentation'.

Line 25: change <1 to < 1.

Line 26: Not sure about the context of this sentence. On which process/property ice-ice collisional breakup has negligible impact? On ice concentration or dynamics?

**Introduction**

Lines 34-35: Citing more recent studies of observational evidences of SIP would be beneficial (e.g., Korolev et al. 2022).

Lines 35-43: Where the term SIP is introduced, please mention the region (mixed-phase) where it mainly occurs in clouds.

Lines 39-41: Cite previous literature (e.g., Lohmann et al.; Kudzotsa et al. 2016; Han et al. 2024; Waman et al. 2025) supporting this.

Lines 41-43: I do not see that the manuscript attempt to improve the representation of SIP processes in numerical model. Rather, the effect of SIP processes is quantified in deep convective clouds using existing parameterizations. Please rephrase.

Line 50: Can the authors comment and acknowledge findings of recent study by Seidel et al. 2024, which see no experimental evidence of rime-splintering, especially in convective conditions. Considering the findings of Seidel et al. 2024, what is the relevance rime-splintering process and its existing parameterization in representing SIP at such warmer subzero levels?

Line 57: Please cite relevant previous studies that used Unified framework to study SIP.

Line 64: Waman et al. 2022 do not explicitly quantify the impact of SIP on the mentioned ice growth processes. Please correct/clarify more.

**Methodology**

Line 113: 'are' instead of 'is'?

How are cloud droplets activated in CASIM? Does the scheme explicitly account for the activity of soluble aerosols as CCN, or is the CCN spectrum prescribed from observations? What is the nature of aerosols (continental/marine)?

Line 115: I believe with Cooper, only immersion mode of heterogeneous ice nucleation is represented. Can the authors clarify how other heterogeneous ice nucleation modes (e.g., deposition), that can be crucial at colder temperatures, are treated in CASIM? Additionally, is homogeneous freezing of aqueous aerosols represented separately from homogeneous droplet freezing, and if not, what are the implications for ice formation at cirrus temperatures?

Line 116: 'rime-splintering' instead of 'riming splintering'?

Line 118: 'other newly implemented SIP processes' instead?

Line 135: What is the value of rime fraction used to represent ice-ice collision in Eq. 3? Also, can the authors comment on how the re-fitted values of the parameters in Eq. 3, given by Grzegorczyk et al. 2023 (Table 3) would influence the predictions from ice-ice collisional breakup?

Eq. 6: $N_T$ should be $N_{M1T}$?

General comment: Overall, the considered SIP processes are described adequately. However, the study does not appear to consider SIP during sublimation of ice particles in subsaturated cloudy environments (e.g., Deshmukh et al., 2022; also see Korolev and Leisner 2020 for limitations). While I understand this mechanism is still under active investigation, it could be relevant in tropical anvil outflow/downdraft regions (Waman et al. 2022). Could the authors briefly comment on the reason for excluding this process and its possible implications for the presented results? Also, adding a table of symbols used would be helpful.

Line 217-218: I do not understand what really makes Hector as an ideal case for studying SIP.

Line 221: Would be nice to mention cloud-base (LCL) and cloud top from Fig. 2.

**Results**

Line 291: space between '(CFAD)' and 'of'.

Line 306: 'SIP' instead of 'secondary ice production'?

Figure 4: For better comparison, is it possible to show isotherms also in (a)? Also show in the form of text '0ºC', '-20ºC', '-40ºC', '-60ºC' (The same can be followed for Figs. 9 and 10).
Figure 5: Also mention date (1 December 2005) in the caption.

Lines 351-356: Can the authors comment on how well the model captures the observed surface precipitation? Overall, I see that the model significantly underpredicts the surface precipitation, both in all-SIP and no-SIP experiments.

Line 356: 'diffused' instead of 'diffuse'?

Line 357: 'The convective core is less pronounced…' This is not clear. How is it interpreted? Additional analysis would be helpful to support this.

Line 358-359: This is also not quite clear. How all-SIP shows a more localized and organized precipitation? Also, I do not agree that the all-SIP experiment resembles the overall observed convective core as the simulations substantially fails to capture the observed precipitation features.

Figure 6d: is this all-SIP minus no-SIP? Mention clearly in the caption.

Line 366: Cite Figure 6d.

Line 367: I do not see these features; can the authors describe this more? How exactly no-SIP shows more evenly distributed precipitation than all-SIP?

Line 369: 'realistic reproduction': Both all-SIP and no-SIP rather captures more localized precipitation events in the simulated domain and not the overall precipitation. Please rewrite as precipitation differ significantly between the simulations and observation.

Line 371: What does 'focus' mean here? How convective rain(fall) is identified?

Line 377: 'increase' instead of 'increases'?

Line 378: Why the mode 2 results in more pronounced convective core?

Lines 366-405: Can the authors explain briefly in the manuscript what possible factors SIP alters that result in the predicted change in precipitation? Although paragraph (lines 405-413) explains the overall influence of a combination of various SIP processes, the exact cause in each case (in Fig. 7) is not discussed.

Lines 457-460: Although mode 2 is less efficient and more confined than mode 1, for what possible reasons does RS+M2 produces more precipitation (Fig. 7e)?

Line 470: 'increase' instead of 'increases'?

Line 473-474: A suggestion: Time-height maps of total ice concentrations in all-SIP and no-SIP, and a similar difference plot (all-SIP minus no-SIP) would be helpful to visualize increased extensiveness over longer period in all-SIP case.

Line 532: Previous work by Qu et al. (2022) and Grzegorczyk et al. (2025) can be cited here.

Line 569-570: This needs more clarification, as both warm and cold rain processes can happen simultaneously at subzero levels.

**References**

Korolev, A., DeMott, P.J., Heckman, I., Wolde, M., Williams, E., Smalley, D.J. and Donovan, M.F., 2022. Observation of secondary ice production in clouds at low temperatures. *Atmospheric Chemistry and Physics*, *22*(19), pp.13103-13113.

Lohmann, U., 2006. Aerosol effects on clouds and climate. *Space Science Reviews*, *125*(1), pp.129-137.

Kudzotsa, I., Phillips, V.T. and Dobbie, S., 2016. Aerosol indirect effects on glaciated clouds. Part 2: Sensitivity tests using solute aerosols. *Quarterly Journal of the Royal Meteorological Society*, *142*(698), pp.1970-1981.

Waman, D., Jadav, A., Patade, S., Gautam, M., Deshmukh, A. and Phillips, V., 2025. Mechanisms for Indirect Effects from Ice Nucleating Particles on Continental Clouds and Radiation. *Journal of the Atmospheric Sciences*.

Seidel, J.S., Kiselev, A.A., Keinert, A., Stratmann, F., Leisner, T. and Hartmann, S., 2024. Secondary ice production–no evidence of efficient rime-splintering mechanism. *Atmospheric Chemistry and Physics*, *24*(9), pp.5247-5263.

Korolev, A. and Leisner, T., 2020. Review of experimental studies of secondary ice production. *Atmospheric Chemistry and Physics*, *20*(20), pp.11767-11797.

Grzegorczyk, P., Yadav, S., Zanger, F., Theis, A., Mitra, S.K., Borrmann, S. and Szakáll, M., 2023. Fragmentation of ice particles: laboratory experiments on graupel–graupel and graupel–snowflake collisions. *Atmospheric Chemistry and Physics*, *23*(20), pp.13505-13521.

Grzegorczyk, P., Wobrock, W., Canzi, A., Niquet, L., Tridon, F., and Planche, C. (2025). Investigating secondary ice production in a deep convective cloud with a 3d bin microphysics model: Part II - effects on the cloud formation and development. Proceedings of the National Academy of Sciences, 314.

Qu, Z., Korolev, A., Milbrandt, J., Heckman, I., Huang, Y., McFarquhar, G., Morrison, H., Wold, M., and Nguyen, C. (2022). The impacts of secondary ice production on microphysics and dynamics in tropical convection. Atmos. Chem. Phys., 22(18):12287–12310.

---

## Author Comment (AC1)

**Responses to reviews**

"Influence of secondary ice formation on tropical deep convective clouds simulated by the

Unified Model"

by Mengyu Sun et al.

Thank you to the editor and reviewers for their time and constructive comments. We have carefully considered all suggestions and addressed them in the revised manuscript. Below, our point-by-point responses are shown in blue, and the corresponding revised sentences are presented in gray italic for clarity.

**Referee #1**

Secondary ice production (SIP) is an uncertain process in different cloud types including the tropical deep convection. Incorporating multiple SIP mechanisms into the CASIM microphysics scheme, the results of a Hector thunderstorm show that SIP greatly increases ice number concentrations and ice water content, expands anvil cloud coverage, and modifies both radiation and rainfall patterns. These results highlight the need to represent SIP in cloud-resolving models to better capture tropical convection and its climatic impacts. As a strong positive, I appreciate the use of small modeling ensembles, which adds robustness to the results. The simulated cases are also well presented and compared with observations. However, the analysis remains somewhat limited and would benefit from greater depth. I recommend strengthening the analysis before the manuscript can be considered for publication.

**Overall comments**

In analyzing the mean values of different cloud properties, have you considered that the simulations may encompass different cloud volumes? Conditional sampling could introduce biases if only mean values are compared. In addition, the choice of modeling framework, along with the level of microphysical detail and spatial resolution, may strongly influence the results. These aspects should be discussed in greater depth.

We thank the referee for the positive assessment and for raising these important points. We agree that both conditional sampling and the model configuration can influence diagnosed properties. In our analysis, we apply a consistent sampling approach across all simulations, and our conclusions are based on the relative SIP-induced differences rather than the absolute values. We also acknowledge that the chosen model resolution and microphysical configuration may affect the quantitative values and contributes to some of the biases relative to observations. Our conclusions focus on the relative SIP-induced differences, which remain robust. Detailed responses and manuscript modifications related to these aspects are provided in the replies to the specific comments below.

**Specific comments**

Line 16: "...including droplet fragmentation (Mode 1 and Mode 2)..." Referring to the modes here is a technical detail that is not widely known. Please either provide a brief explanation of these modes or omit mentioning them from the Abstract.

We have removed the references to "Mode 1 and Mode 2" from the Abstract to avoid unnecessary technical detail. The sentence now reads (see line 15): "...including droplet fragmentation..."

Lines 74–75: Is the effect limited to increased ice loading, or is it possible that the anvil also extends over a larger area?

We have clarified that the reduction in OLR is not only related to increased ice loading, but also to the expansion of anvil cloud area. The revised text now reads (see lines 81–82):

"Increased ice loading and the expansion of anvil cloud area may lower outgoing longwave radiation (OLR) due to colder and more extensive cloud tops, ..."

Section 2.2: A number of equations are presented, many of which appear to originate from earlier publications. If these are identical to previous formulations, please explain why they are repeated here. There is also an option to move these into supplementary material.

Yes, we agree that many of the equations originate from previous studies. We reproduced them to explicitly show the parameterizations in our simulations. This ensures clarity and transparency of the modelling framework. We therefore prefer to retain them in the main text. The following sentence has been added (see lines 136–137):

"The corresponding equations are reproduced below to explicitly show the parameterizations applied in our simulations."

Lines 306–307: "...inclusion of secondary ice production leads to a reduction in mean reflectivity values in middle levels. This is mainly attributed to the smaller ice particles aloft." What exactly does "mainly" mean here? Which hydrometeor categories are responsible for the change? Do all categories show decreased size and increased number, or is the effect limited to specific ones? Even if SIP generates new particles, those originating from primary freezing should still grow almost as fast as those without SIP if ice—ice collisional breakup is inefficient. Please clarify.

We thank the reviewer for this helpful comment. We have revised the text to avoid ambiguity and specified the hydrometeor categories responsible. The reduction in mean reflectivity at middle levels is associated with the presence of more numerous small ice-phase particles aloft under SIP, which shifts the overall size distribution toward smaller diameters and lowers bulk reflectivity ( $Z \propto D^6$ ). We also note that ice particles formed by primary freezing continue to grow at comparable rates when ice—ice collisional breakup is inefficient, and the SIP-generated particles shift the overall size distribution, leading to weaker reflectivity. Furthermore, we added clarification in the analysis of Fig. 9, showing that the increase is largest for cloud-ice crystals, while changes in graupel are weaker and snow shows little sensitivity.

**Revised text (Section 3.2, lines 331–336):**

"It demonstrates that the inclusion of SIP leads to a reduction in mean reflectivity values in middle levels, which can be explained by the presence of more numerous small ice-phase particles aloft. Ice particles formed by primary freezing are expected to grow at comparable rates when ice—ice collisional breakup is inefficient, and the additional SIP-generated ice particles shift the overall size distribution toward smaller diameters, leading to weaker reflectivity."

**Revised text (Section 3.3.1, lines 521–523):**

"..., the reflectivity decrease in Figure 4 is mainly associated with the increase in small ice crystal number; graupel contributes minor changes relative to crystals, whereas snow remains weakly sensitive to SIP."

Lines 312–314: "Below 3 km, the model underestimates the frequency of reflectivity values exceeding 5 dBZ..." Does this mean that raindrops are evaporating too rapidly or are not large enough in the beginning, or could the discrepancy also stem from how reflectivity is calculated and conditionally sampled?

We clarified the likely causes of the bias. The text now states that the underestimation likely reflects uncertainties in warm-rain processes (e.g., cloud-to-rain autoconversion, accretional growth, rain evaporation, and assumed drop-size distributions), and that differences between the model reflectivity diagnostic and the observational product near the melting layer may also contribute. We also note that this bias is similar in the SIP and no-SIP experiments and does not affect the SIP-related conclusions. The revised sentence reads (lines 341–346):

"This likely reflects uncertainties in the representation of warm-rain processes, including cloud-to-rain autoconversion, accretion of cloud water by rain, rain evaporation, and the assumed particle size distributions. Differences between the model reflectivity diagnostic and the observational product near the melting layer could also contribute to the underestimation. The bias is similar in the SIP and no-SIP experiments and therefore does not affect the SIP-related conclusions."

Lines 356–359: "The convective core is less pronounced..." Is it possible to get some statistics to support this. Visually this is not too evident.

To avoid ambiguity, we now define the heavy-rainfall region as the area with accumulated rainfall  $\geq$  40 mm (e.g., Grzegorczyk et al., 2025b) and overlay the corresponding contours in Figures 6 and 7. We verified that using nearby thresholds (30–50 mm) yields the same qualitative contrast (not shown). Specifically, the heavy-rainfall region indicates that the no-SIP run produces a broader, more diffuse precipitation with rainfall spread over a wider area (Fig. 6c). The all-SIP run exhibits a more localized and compact precipitation pattern that better captures the observations (Fig. 6b). We have modified the text as follows (lines 387–392):

"The no-SIP simulation produces a broader and more diffuse precipitation field, with rainfall spread over a wider area (Figure 6c). For spatial diagnostics, we define a heavy-rainfall region as the area with accumulated rainfall ≥40 mm (black contours in Figures 6b, c; Grzegorczyk et al., 2025b). This heavy-rainfall region is more spread out in the no-SIP than in the all-SIP case. In contrast, the all-SIP simulation shows a more localized and compact precipitation field that better matches the observed spatial pattern (Figure 6b), ..."

Figure 8: How large is the uncertainty in the "observed" precipitation data?

The observed precipitation is from the Darwin C-band polarimetric radar (CPOL). CPOL-based quantitative precipitation estimates are considered reliable for tropical rainfall, with residual uncertainties arising mainly from reflectivity/differential-reflectivity calibration and C-band attenuation, as documented for Darwin in Louf et al. (2019) and Jackson et al. (2021). We now clarify this in the paper in Data (Section 2.3) and Results sections. Our analysis focuses on relative contrasts between SIP and no-SIP runs, which remain robust across sensitivity tests. We have modified the text as follows:

Data (lines 259–262): "CPOL-based quantitative precipitation estimates are considered reliable for tropical rainfall, with residual uncertainties arising from reflectivity/differential-reflectivity calibration and C-band attenuation (Louf et al., 2019; Jackson et al., 2021)."

Results (lines 395–397): "On the observational side, CPOL-based precipitation data also carry documented uncertainties (see Section 2.3). Nevertheless, our analysis focuses on the contrasts between SIP and no-SIP runs, which remain robust across the sensitivity tests."

Lines 485–486: "Ice–ice collisional breakup remains largely insignificant..." Why is the discussion restricted primarily to graupel? The process involves all ice hydrometeors. Given that the all-SIP simulation produces the highest ice concentrations, how can you rule out a contribution from collisional breakup after other processes increase concentrations?

We have now clarified why graupel is emphasised and how we assess the contribution of collisional breakup.

Why the discussion focuses on graupel. In our implementation of the Phillips et al. (2017b) breakup scheme (Sec. 2.2.2), the BR source depends on (i) a collision-frequency term and (ii) a fragments-per-collision term that increases with collision kinetic energy and riming intensity. Numerical simulations with the same scheme indicate that snow–graupel collisions account for the majority of breakup fragments under convective conditions (Phillips et al., 2017b). We have made this explicit in Sec. 2.2.2 (lines 175–176):

"Numerical simulations indicate that snow–graupel collisions account for the majority of breakup fragments under convective conditions (Phillips et al., 2017b)."

How we evaluate BR in this case. Diagnostics show that snow number varies little among the experiments (Fig. 9d), so breakup associated with snow is of similar magnitude across runs and cannot explain the inter-experiment contrasts in ice number. At the same time, graupel concentrations are low through the column (~1–10 L-1 near 8 km; Fig. 9b), which limits the collisions that are effective for breakup in this scheme. Consistent with these conditions, the run with breakup enabled and the reference run have very similar ice-number profiles (Figs. 9a, e). We therefore conclude that, for this case, BR contributes little relative to the droplet fragmentation, while acknowledging that a minor additional BR contribution may arise once concentrations are elevated by other SIP processess. We have now clarified in the revised version (lines 489–493, lines 509–514):

"..., the RS+BR experiment (including ice—ice collisional breakup) differs little from the RS-only case across the column: the profiles are nearly identical and fall within the ensemble spread (Figures 9a, e). This indicates a limited contribution from breakup under the present conditions. A further contribution may arise once concentrations are elevated by other SIP processes."

"Because snow varies little across experiments, snow-involved breakup is likely similar in

magnitude across runs and cannot explain the contrasts in ice number. Meanwhile, graupel concentrations are low throughout the column, which limits the collisions effective for breakup in this scheme. Consistent with this, the simulation with breakup enabled (RS+BR) and the RS-only run show similar ice-number profiles (Figures 9a, e)."

Figure 10: Is the amount of snow reduced because of SIP? Not evident from the ice particle concentrations in Figure 9. Can this affect the outcome, for example related to changes in reflectivity?

In our simulations, we do not see clear indications that SIP reduces snow. Snow number concentration shows only weak sensitivity across experiments (Fig. 9d), and when snow is included the column IWC increases in all simulations, with the largest enhancement above  $^{\circ}$ 9 km where snow contributes most to the mass budget (Fig. 10b). These suggest that SIP-related changes in snow are small compared to ice crystals under the present conditions. The decrease in mean reflectivity aloft (Fig. 4b) is consistent with a SIP-related shift toward more numerous small ice crystals, which affects the particle-size distribution (Z  $\propto$  D6). We have clarified this in the text (lines 549–552, 555–559).

"When snow is included (Figure 10b), column IWC increases across all simulations, with the largest enhancements above  $^{\circ}9$  km ( $T \approx -20$   $^{\circ}C$ ) where snow dominates the mass budget. In this layer, snow is both more numerous than graupel (Figures 9b, d) and larger in characteristic size than ice crystals, so it contributes a greater share of IWC. Once snow is accounted for, the inter-experiment differences that mainly arise from crystals are partly diluted (Figures 10a, b), as graupel remains limited.

"These patterns are consistent with the number concentration profiles (Figures 9d, e). SIP substantially increases ice crystal numbers (and to a lesser extent graupel) in the upper troposphere and anvil, whereas snow shows weak sensitivity to SIP. Consequently, adding snow reduces the contrast among experiments, yet the SIP-driven shift toward more numerous small crystals remains evident, explaining the decrease in mean reflectivity aloft (Figure 4b)."

Figure 10 (Maximum updraft velocity): Could you compare the simulated maximum updraft velocities with those from higher-resolution studies, e.g., large-eddy simulations by Dauhut et al. (2015)? The values here appear smaller than might be expected, which could have implications for SIP efficiency and overall precipitation formation.

The smaller maximum updrafts in Fig. 10c relative to higher-resolution LES (e.g., Dauhut et al., 2015) may result from several resolution-related factors. At km-scale grid spacing (1.5 km), processes such as numerical diffusion and subgrid-scale parameterizations can weaken narrow, intense updrafts, leading to lower peak vertical velocities than in 100 m LES. Nevertheless, our sensitivity tests demonstrate that the SIP—no-SIP contrasts and the overall precipitation response remain robust. A note has been added in the revised manuscript to clarify this point (lines 570–574):

"Given the km-scale grid (1.5 km) used here, the peak vertical velocities are smaller than in higher-resolution large-eddy simulations (e.g.,  $\sim$ 100 m; Dauhut et al., 2015). This likely reflects resolution effects that weaken intense updrafts, for example through numerical diffusion and subgrid-scale treatment. The contrasts between SIP and no-SIP simulations remain robust, as demonstrated by the sensitivity experiments."

Lines 569–570: "SIP diverts condensate away from warm-rain processes into less efficient ice-phase pathways." Please clarify this statement. Ice particles typically grow faster than liquid droplets, so in what sense is the pathway "less efficient"?

In this context, "less efficient" refers not to the growth rate of ice particles but to the overall conversion efficiency into surface precipitation. SIP produces numerous ice particles that are lofted to upper levels. This makes the ice-phase pathway less efficient in delivering condensate to surface rainfall compared to the warm-rain processes. We have clarified this wording in the revised version (lines 610–612).

"...suggesting that in the mixed-phase region and above, SIP shifts more condensate from warm-rain processes toward ice-phase pathways, which are less efficient at producing surface precipitation."

Line 578: "...the updraft velocity (Wmax) increases by ~10%... and does not substantially intensify peak convection. A 10% change in cloud dynamics due to microphysical processes could be considered significant. What threshold would you regard as "substantial"?

We agree that a ~10% increase in Wmax is a measurable change. Our intention was not to downplay its magnitude but to emphasize that, in this case, the increase had limited impact on the peak rainfall rate. We have revised the text to clarify this as (lines 632–633):

"..., the updraft velocity ( $W_{max}$ ) increases by ~10% (Figure 10c), which is measurable but has limited impact on peak rainfall metrics in this case."

**References**

Dauhut, T., Chaboureau, J.-P., Escobar, J. and Mascart, P. (2015), Large-eddy simulations of Hector the convector making the stratosphere wetter. Atmos. Sci. Lett., 16: 135-140. https://doi.org/10.1002/asl2.534

**Referee #2**

The paper numerically investigates the possible role of secondary ice production (SIP) in forming overall ice number concentration, and its influence on cloud properties in a convective case observed during ACTIVE campaign in early December 2005. For this, the authors incorporated three SIP processes (Hallett-Mossop rime-splintering, ice-ice collision, breakup of freezing raindrop) in UK Met Office Unified Model's 2-moment CASIM microphysics scheme. The study finds that, through increased ice number and mass in the upper region, SIP can modify the anvil structure in the simulated thunderstorm, and changes the precipitation formation, especially associated with the convective core. The ensemble simulations are also performed that illustrates the robustness of the presented results. The introduction is comprehensive, and sufficiently discusses the recent advancements and challenges in SIP research, and the methodology is sound. The simulated properties, such as the radar reflectivity, OLR, and precipitation are compared and validated against the observations. While the overall presentation is good, the paper could benefit from a more detailed discussion and comparison of how the simulated microphysical properties agrees well with

the observations. Nevertheless, the study addresses a timely and important topic and may be considered for publication after satisfactorily addressing the following concerns.

**General comments**

More details of radiation and microphysics scheme used are needed. The presented validation is reasonable but could have benefited from additional comparison, such as observed liquid/ice properties, if such observations (from satellite or other platforms) are available. Also, many findings presented in the manuscript can be supported by some previous studies, and can also be acknowledged in the introduction section to strengthen the proposed research questions.

We thank the referee for these constructive suggestions. We agree that clearer documentation of the radiation and microphysics schemes, and a stronger connection to previous studies, are important. In the revised manuscript, we provide additional detail on the radiation scheme and summarize the SIP parameterizations implemented in our simulations. For this Hector case, no additional co-located observations of liquid/ice properties are available beyond the datasets already used. We agree that microphysical evaluation with richer observations will be an important direction for future work. We have also expanded the discussion of relevant previous studies to better situate our results.

Detailed responses and the corresponding manuscript changes related to these points are provided in the replies to the specific comments below.

**Specific comments**

**Abstract**

Line 14: Since the study mainly quantifies the impacts of SIP on the simulated clouds without attempting to modify/improve the existing SIP parameterizations (beyond the use of revised  $\Phi$  from James et al. 2021), I would suggest rephrasing this sentence, as the focus is not on quantifying and reducing uncertainties in the modelled SIP processes.

Thank you for the suggestion. We have revised the sentence to more accurately reflect the focus of the study (lines 14–15):

"Secondary ice production (SIP) plays an important role in tropical deep convection. This study implements multiple SIP mechanisms, including droplet fragmentation and ice—ice collisional breakup, ..."

Line 16: Mode 1 and 2 of drop shattering are scheme-specific terms and are not widely recognized. These are proposed by Phillips et al. (2018) to represent drop shattering in collision between rain/drizzle drop with ice particle. Other schemes of SIP in drop shattering (e.g., Sullivan et al. 2018) only consider shattering of raindrop during freezing, initiated due to immersed INP, without separating mode 1 and 2. Better to omit using mode 1 and mode 2 and say only 'drop fragmentation'.

We agree and have revised the sentence to use "droplet fragmentation" instead.

Line 25: change

**Figure S1.** Time–height plots of ice water content (IWC; g m-3) for simulations: (a) all-SIP, (b) no-SIP, and the difference between allSIP and noSIP simulations (i.e., allSIP minus noSIP). Panels (a–c) are averaged over regions where IWC > 0.01 g m-3. The 0, -20, -40, and -60 °C isotherms are shown by the dashed lines.

Line 532: Previous work by Qu et al. (2022) and Grzegorczyk et al. (2025) can be cited here. We have added citations to Qu et al. (2022) and Grzegorczyk et al. (2025).

Line 569-570: This needs more clarification, as both warm and cold rain processes can happen simultaneously at subzero levels.

We agree that at subzero temperatures warm-rain and ice-phase processes can happen simultaneously. We didn't to intend to imply a complete replacement of warm-rain processes by ice, but a shift in condensate partitioning toward ice-phase pathways. To clarify, we have revised the text to (lines 610–612):

"..., suggesting that in the mixed-phase region and above, SIP shifts more condensate from warm-rain processes toward ice-phase pathways, which are less efficient at producing surface precipitation."

**Pierre Grzegorczyk**

This study investigates the influence of secondary ice production (SIP) on a tropical deep convective cloud (Hector) using the 2-moment microphysics scheme (CASIM) of the UK Met Office Unified model. The paper highlights significant effect of SIP on important cloud properties such as the ice particles number concentration, ice water content, precipitation amount and repartition as well as radiative properties of the cloud system. Overall, the paper

is well written, and I enjoyed reading it. The results bring interesting and important results about the importance of SIP mechanisms. I especially liked the ensemble simulation performed in the study which gives even more robustness to the numerical sensitivity tests. I have a few questions, remarks and suggestions about the results and their interpretation. I think that my comments are more suggestive and can considered to be relatively minor. The current version is, in my opinion, almost ready for a final publication even if some specific points can be strengthened and clarified.

Thank you, Dr Grzegorczyk for your review. We respond to your questions and suggestions below.

**Comments:**

Line 51-52: I would argue the opposite, supercooled droplets and graupel are common in convective updrafts which favor strong supersaturations and transport supercooled liquid droplets. Evidence of riming in convective regions can be found, for example, in https://doi.org/10.1175/JAS-D-25-0021.1

We agree that supercooled droplets and graupel can indeed be present in deep convections, providing favorable conditions for the HM process. We have revised the text to reflect that (lines 55–56):

"These conditions may occur under favorable thermodynamic environments (Bazantay et al., 2025), but are not always present in tropical convective updrafts (Field et al., 2017; Huang et al., 2022)."

Intensification of strong precipitation (Line 65-67 and Section 3.2): Our studies (Grzegorczyk et al., 2025b and Grzegorczyk et 2025c, https://doi.org/10.5194/acp-25-10403-2025) show that for two types of convective clouds strong precipitation (>40 mm) is especially reduced by SIP. It may be interesting to mention this to balance your results and those of Sullivan et al. (2018) in the results and discussion section?

We thank the reviewer for this suggestion. To provide a balanced perspective, we have added brief statements in the Results and Discussion sections that acknowledge differing findings (Grzegorczyk et al., 2025b, c; Sullivan et al., 2018):

**Results (Section 3.2, lines 453-455):**

"It is worth noting that previous studies have reported varying impacts of SIP on precipitation intensity. For example, Grzegorczyk et al. (2025b, c) found that SIP can reduce heavy rainfall (>40 mm), whereas Sullivan et al. (2018) reported localized enhancement within convective regions."

**Discussion (lines 634–636):**

"Previous studies reported varied impacts of SIP on precipitation intensity: reductions in heavy rainfall (e.g., >40 mm) have been documented (Grzegorczyk et al., 2025b, c), whereas localized enhancements within convective regions have also been reported (Sullivan et al., 2018)."

Line 131: Is the riming rate obtained in a similar way as for ice-ice collision breakup (Eq. 2) and drop shattering (Eq. 8)? Since HM and DS both depend on collisions between ice particles and drops, do you consider that HM and DS can occur simultaneously when a certain

number/mass of drops are freezing by collisions with ice particles?

The riming rate is not obtained in the same way as ice—ice collisional breakup (Eq. 2) or droplet shattering (Eq. 8). In CASIM, riming rate is computed with a sweepout (collection) formulation in which the ice/snow collector sweeps cloud water, yielding a single-integral term (Field et al., 2023, Appendix A, Eq. A12).

$$P_{\text{xac }y} = \frac{\pi n_x a_x \Gamma(3 + b_x + \mu_x) E_{xy} q_y}{4(\lambda_x)^{3 + b_x + \mu_x}} \left(\frac{\rho_0}{\rho}\right)^{g_x}, \quad (A12)$$

where nx is the number concentration mixing ratio of the collector hydrometeor x, qy is the mass mixing ratio of the species being collected, Exy is the collection efficiency, ax, bx,  $\mu$ x, and gx are the parameters for the collector hydrometeor x as defined in Table A1, and  $\rho$ 0 = 1.22 kg·m-3 is a reference value of air density.

When both collectors and collected species have non-negligible fall speeds, CASIM uses a binary-collection formulation (Appendix A, A13).

$$P_{xac y} \approx c_{y} \delta E_{xy} \left[ \frac{\Gamma(1 + \mu_{y} + d_{y})}{\lambda_{y}^{1 + \mu_{y} + d_{y}}} \frac{\Gamma(3 + \mu_{x})}{\lambda_{x}^{3 + \mu_{x}}} + 2 \frac{\Gamma(2 + \mu_{y} + d_{y})}{\lambda_{y}^{2 + \mu_{y} + d_{y}}} \frac{\Gamma(2 + \mu_{x})}{\lambda_{x}^{2 + \mu_{x}}} + \frac{\Gamma(3 + \mu_{y} + d_{y})}{\lambda_{y}^{3 + \mu_{y} + d_{y}}} \frac{\Gamma(1 + \mu_{x})}{\lambda_{x}^{1 + \mu_{x}}} \right],$$
(A13)

where  $\delta Vxy$  is the larger of the difference in mass-weighted fall speeds. Droplet shattering (DS) involves rain–ice collisions, so its rate is evaluated in the binary-collection framework. Hallett–Mossop (HM) is tied to graupel/snow riming of cloud droplets within the temperature range (~-3 to -8 °C), whereas DS arises from freezing and breakup of supercooled raindrops during rain–ice collisions. In our setup, HM and DS can operate simultaneously if their respective criteria are met within the same grid and time step.

Paragraph 2.2.3: It is a good point to describe the implementation of SIP in CASIM in detail. I am just wondering what values were considered for the ice-ice and ice-drop collision efficiencies? Additionally, how are the terminal velocities of ice, snow, and graupel calculated? I think it is important since it directly determines the rate of SIP processes.

In our CASIM setup, ice—ice and ice—drop collision efficiencies follow the scheme defaults documented in Field et al. (2023), Appendix A, Table A3. Terminal fall speeds for ice, snow, and graupel use parameters from Table A1.

We have added the following sentence in the revised manuscript (lines 177–178):

", the collection efficiencies and terminal fall-speed parameters are given in Field et al. (2023), Appendix A, Tables A3 and A1, respectively."

TABLE A3 Values of collection efficiencies  $E_{xy}$  with species x collecting species y

| Collection efficiency | Value                                | Routine               | Reference                                         |
|-----------------------|--------------------------------------|-----------------------|---------------------------------------------------|
| $E_{ m iw}$           | 0                                    | src/ice_accretion.F90 |                                                   |
| $E_{ m sw}$           | 0.5                                  | src/ice_accretion.F90 | Following snow riming in Furtado and Field (2017) |
| $E_{ m gw}$           | 1.0                                  | src/ice_accretion.F90 |                                                   |
| $E_{ m sr}$           | 1                                    | src/ice_accretion.F90 |                                                   |
| $E_{ m gr}$           | Inactive                             |                       |                                                   |
| $E_{ m ri}$           | 1                                    | src/ice_accretion.F90 |                                                   |
| $E_{ m si}$           | $0.2e^{0.08T_c}$                     | src/ice_accretion.F90 |                                                   |
| $E_{ m gi}$           | $0.2\mathrm{e}^{0.08T_{\mathrm{c}}}$ | src/ice_accretion.F90 |                                                   |
| $E_{ m gs}$           | $0.2\mathrm{e}^{0.08T_\mathrm{c}}$   | src/ice_accretion.F90 |                                                   |
| $E_{ m ss}$           | $0.1e^{0.08T_{\rm c}}$               | src/aggregation.F90   |                                                   |
| $E_{ m rr}$           | 1.0                                  | src/aggregation.F90   | Beheng (1994)                                     |
| $E_{\rm ii}$          | Inactive                             |                       |                                                   |
| $E_{ m gg}$           | Inactive                             |                       |                                                   |

Note:  $T_c$  is the temperature in Celsius.

TABLE A1 CASIM hydrometeor parameters

|         | Terminal fall speeda M                        |                     |                  | Mass-di                                        | Mass-dimensionb |   |                     | Shape parameter |                    |
|---------|-----------------------------------------------|---------------------|------------------|------------------------------------------------|-----------------|---|---------------------|-----------------|--------------------|
| Species | а                                             | b                   | $\boldsymbol{f}$ | Note                                           | c               | d | Note                | mu              | Note               |
| Cloud   | $3 \times 10^7$                               | 2                   | 0.5              | Stokes sphere                                  | 522             | 3 | Liquid sphere       | 2.5             |                    |
| Rain    | a 1 =4854,
a 2 =-446 | b1=1.0,
b2=0.782 | 0.5              | Abel and Ship-
way 2007,
g1=0,g2=4085.35 | 522             | 3 | Liquid sphere       | 2.5             |                    |
| Ice     | $6 \times 10^6$                               | 2                   | 0.5              | Stokes sphere                                  | $200\pi/6$      | 3 | Sphere              | 2.5             |                    |
| Snow    | 12                                            | 0.5                 | 0.5              |                                                | 0.026           | 2 | Cotton et al., 2013 | 2               | Field et al., 2007 |
| Graupel | 253                                           | 0.734               | 0.422            |                                                | $500 \pi/6$     | 3 |                     | 2.5             |                    |

 $^a v = a D^b (\rho_0/\rho)^f$ , SI units, D is particle maximum span and  $\rho_0$  is the reference density of air (1.22 kg m-3). For Abel and Shipway rain  $v = [a_1 D^{b1} e^{(\cdot g1D)} + a_2 D^{b2} e^{(\cdot g2D)}] (\rho 0/\rho)^f$

Line 259-260 and lines 583-596: Great idea to use four ensemble members to make the results more robust.

Thank you for the comment.

Line 299-304 for Fig 4: The cloud top altitude seems to be lower in the no SIP case which is quite interesting as the same results were obtained in Qu et al. (2022) (https://doi.org/10.5194/acp-22-12287-2022) while we found the opposite in our study (Grzegorczyk et al. 2025b).

In our Hector case, SIP produces a higher cloud top and closer agreement with observations. We have discussed this in the revised version (lines 326–330):

"In this case, SIP yields a higher cloud top and closer agreement with observations, consistent with previous simulations (e.g., Qu et al., 2022). Cloud top responses can vary across configurations. For example, an idealized single cloud study reported lower tops when SIP produced small ice that depleted cloud droplets and limited upper-level ice formation (Grzegorczyk et al., 2025b)."

Fig 5: I am not a specialist in radiative transfer, but do you think that the changes in OLR and

 $^{b}m = cD^{d}$ , SI units.

OSR due to SIP can be important at larger scale for the climate?

In our Hector case, SIP may influence larger-scale radiative budgets via changes in anvil properties. However, assessing climate relevance would require statistics across many events and environments. We have added the following sentence in the Discussion section (lines 602–604):

"..., potentially influencing regional and large-scale climate feedback. Further work evaluating these effects across different events and environments, and using long integrations with parameterized SIP, would be helpful for quantifying their climate relevance."

Line 406-409: Our study (Grzegorczyk et al (2025c) (https://doi.org/10.5194/acp-25-10403-2025) as well as the one of Dedekind et al. (2021) (https://doi.org/10.5194/acp-21-15115-2021) investigated the reasons for the SIP influence on ground precipitation, which may support your statement.

We have now cited these studies in the revised manuscript.

Fig. 9b: The graupel concentration (1-10 L-1) seems high, and close to ice and snow concentration below 8 km. Even if I am not familiar with the number of graupels in models, I am unsure if this is realistic. Regarding lines 486 and 618-619 I am not sure that the reason for the weak ice-ice breakup is due to the lack of graupel collisions since their concentration look relatively high. Could the weak ice-ice breakup result from snow-snow or graupel-snow collisions?

The horizontally averaged graupel number below 8 km (≈1–10 L-1) is on the high side of aircraft-based expectations for tropical convection (e.g., Lasher-Trapp et al., 2016), but falls within the spread reported by convection-permitting studies (e.g., Phillips et al., 2017b). Regarding the ice—ice breakup (BR), the breakup rate depends on the collision kernel (e.g., collisional kinetic energy, CKE) rather than graupel number alone (Phillips et al., 2017b). In this Hector case, under a modest IWC, the relatively higher graupel number implies smaller mean graupel size and lower fall speeds, which reduces CKE in graupel—involved collisions. Based on Phillips et al (2017b), these conditions keep the overall BR weak.

We have modified the corresponding analysis in the revised manuscript (lines 493–498, 668–674):

"..., graupel number concentrations below 8 km (~1–10 L–1) fall within the spread reported by convection-permitting studies (e.g., Phillips et al., 2017b), but are on the high side of aircraft-based observations for tropical convection (e.g., Lasher-Trapp et al., 2016). Higher graupel number generally implies smaller mean size and reduced fall-speed differences, which lowers the graupel-involved collisional kinetic energy and likely explains the limited breakup in this case (Phillips et al., 2017b)."

"BR remains weak because graupel-involved collision energetics are limited under warm profiles. At comparable IWC, a higher graupel number implies smaller mean size and weaker mass-weighted fall-speed contrasts, which reduces the collisional kinetic energy that controls breakup efficiency as formulated by Phillips et al. (2017b). Numerical experiments suggest that graupel—snow interactions provide the major source of breakup fragments under convective conditions (Phillips et al., 2017b). Therefore, the overall BR contribution remains small when graupel-involved energetics are limited."

Fig 9e: The total increase in ice particle number with SIP seems relatively small (less than one order of magnitude). Is this concentration realistic for a deep convective cloud? The production of ice particles by SIP in CASIM could be maybe validated against in situ aircraft observations in a future study?

Figure 9e shows horizontally averaged vertical profiles over a multi-hour window, not instantaneous values. Such temporal and spatial averaging could naturally smooth local peaks, so the net increase appears smaller than an order of magnitude. SIP produces numerous small ice particles aloft, but subsequent sedimentation and microphysical processing (e.g., depositional growth, aggregation/riming, sublimation) could reduce number contrasts. Hence the time-average response is moderate even relative to local, instantaneous enhancements. We agree that validation against in-situ aircraft observations would be valuable. For the present Hector case, we do not have collocated in-situ measurements suitable for a one-to-one evaluation of SIP-generated ice number. We view this as a promising direction for future work.

Fig 10: I think that it is important to plot the liquid water content and to see the effect of SIP on it. It can further explain differences in OSR of OLR as well as explain why M1 or M2 are strong.

We have added time—height cross-sections of rain water content (RWC) for the all-SIP and no-SIP runs, together with the difference (all-SIP minus no-SIP), as Supplementary Figure S2. RWC is a more direct indicator of the warm-rain process, which is closely relevant to the precipitation response. In Figure S2, the all-SIP run shows an overall reduced RWC, with the largest reductions above ~5 km and near the surface, consistent with a shift from warm-rain to ice-phase processes. Accordingly, the domain-mean precipitation is lower in all-SIP (Figure 8a).

In this case, changes in OLR are mainly linked to the anvil ice amount and effective emission level. Increased IWC (and smaller ice size) raises the IR optical depth, lifting the effective emission level to colder temperatures and reducing OLR. OSR is sensitive to particle size and phase. Increases in small ice amounts brighten the anvil and raise OSR. (A detailed explanation of why M1/M2 can yield different surface precipitation is provided in the Discussion, Section 4.2; see also our Response to Reviewer #2.)

**Figure S2**. Same as Figure S1, but for rain water content (RWC; g m-3). RWC values are averaged over the area where either the ice water path or rain water path exceeds 1 g m-2.

Lines 612-613 and 656-657: In the original Phillips et al. (2017) formulation, ice production from snow breakup is weaker than from graupel collisions. In my opinion snow breakup is underestimated, as snow particles are more fragile than rimed ones. In Grzegorczyk et al. (2025c), I used different values (see the appendix of the paper) based on earlier experiments (Grzegorczyk et al., 2023) for Phillips et al. (2017) formulation. Testing these new values may be beyond the scope of this paper but it would be interesting to see whether it affects your results.

Thank you for the suggestion. In our CASIM setup we retain the breakup formulation of Phillips et al. (2017) with the Table 1 parameters. We do so because the scheme: (i) incorporates the observed temperature dependence of fragment production based on laboratory and field evidence, and (ii) covers multiple collision types (snow–snow, graupel–graupel, snow–graupel) within a single, energy-conserving framework linked to collisional kinetic energy (CKE).

By contrast, the coefficients proposed in Grzegorczyk et al. (2025c), derived from the earlier laboratory study (Grzegorczyk et al., 2023), were obtained at -15 °C under high supersaturation and subsequently applied to a mid-latitude heavy-precipitation case. As a result, they may yield higher predicted fragment numbers than from Phillips et al. (2017) and being single-temperature fits. Their applicability to warm-profile tropical convection remains to be assessed. Exploring this alternative set in our configuration would be valuable, but is beyond the scope of the present study; we view it as a natural extension for future sensitivity tests.

**Minor comments:**

Line 37: The recent study of Seidel et al. (2024) (https://doi.org/10.5194/acp-24-5247-2024) can be cited.

The study of Seidel et al. (2024) has now been cited where rime-splintering is discussed (lines 57–59).

"Moreover, recent laboratory studies found no experimental evidence of efficient rimesplintering under convective conditions (Seidel et al., 2024), suggesting that the relevance of this process and its parameterization remain uncertain."

Line 39: Additional recent papers about ice-ice breakup can also be included: Yadav et al. (2025) (https://doi.org/10.5194/acp-25-8671-2025) and Gautam et al. (2024) (https://doi.org/10.1175/JAS-D-23-0122.1)

We have now included these recent studies.

Line 41-43: Some references could be used to support that.

We have added supporting references, following the suggestion of Reviewer #2 (lines 43–44). "..., and radiative fluxes at the top of the atmosphere (Lohmann, 2006; Kudzotsa et al., 2016; Han et al., 2024; Waman et al., 2025)."

Line 85: Some references can maybe be cited for the 'Hector-type' cloud.

We have now cited Crook (2001) and Connolly et al. (2006), and a more detailed description is provided in Section 2.3.

Line 164-165: I think that the structure of the sentence could be improved to be clearer We have rephrased it as follows to improve clarity (lines 183–184):

[revised manuscript text omitted]